

# Distribution and cycling of terrigenous dissolved organic carbon in peatland-draining rivers and coastal waters of Sarawak, Borneo

Patrick Martin[1], Nagur Cherukuru[2], Ashleen S. Y. Tan[1], Nivedita Sanwlani[1], Aazani Mujahid[3], Moritz Müller[4]

5   [1]Asian School of the Environment, Nanyang Technological University, Singapore 639798, Singapore
[2]CSIRO Oceans and Atmosphere Flagship, Canberra ACT 2601, Australia
[3]Department of Aquatic Science, Faculty of Resource Science & Technology, University Malaysia Sarawak, 94300 Kota Samarahan, Sarawak, Malaysia
[4]Swinburne University of Technology, Faculty of Engineering, Computing and Science, 93350 Kuching, Sarawak, Malaysia

10   *Correspondence to*: Patrick Martin (pmartin@ntu.edu.sg)



**Abstract.** South-East Asia is home to one of the world's largest stores of tropical peatland, and accounts for roughly 10% of the global land-to-sea dissolved organic carbon (DOC) flux. We present the first-ever seasonally-resolved measurements of DOC concentration and chromophoric dissolved organic matter (CDOM) spectra for six peatland-draining rivers and coastal waters in Sarawak, north-western Borneo. The rivers differed substantially in DOC concentration, ranging from 120–250

5    µmol L$^{-1}$ (Rajang river) to 3,100–4,400 µmol L$^{-1}$ (Maludam river). All rivers carried high CDOM concentrations, with $a_{350}$ in the four blackwater rivers between 70–210 m$^{-1}$, and 4–12 m$^{-1}$ in the other two rivers. DOC and CDOM showed conservative mixing with seawater except in the largest river (the Rajang), where DOC concentrations in the estuary were elevated, most likely due to inputs from the extensive peatlands within the Rajang delta. Seasonal variation was moderate and inconsistent between rivers. However, during the rainier north-east monsoon, all marine stations in the western part of our study area had

10   higher DOC concentrations and lower CDOM spectral slopes, indicating a greater proportion of terrigenous DOM in coastal waters. Photo-degradation experiments revealed that riverine DOC and CDOM in Sarawak are photo-labile: up to 25% of riverine DOC was lost within five days of exposure to natural sunlight, and the spectral slopes of photo-bleached CDOM resembled those of our marine samples. We conclude that coastal waters of Sarawak receive large inputs of terrigenous DOC that is only minimally altered during estuarine transport, and that any biogeochemical processing must therefore occur

15   mostly at sea. It is likely that photo-degradation plays an important role in the degradation of terrigenous DOC in these waters.



## 1 Introduction

The annual flux of terrigenous dissolved organic carbon (tDOC) from rivers into the sea is an important part of the global carbon cycle, estimated as around 0.2 Pg C y$^{-1}$ (Dai et al., 2012). South-East Asian rivers contribute roughly 10% of this flux (Baum et al., 2007; Huang et al., 2017; Moore et al., 2011), chiefly owing to the extensive peat deposits along the coasts of

Borneo and Sumatra (Dommain et al., 2014; Page et al., 2011). The rivers draining these peatlands typically carry milli-molar concentrations of DOC, and are often called "blackwater" rivers (Alkhatib et al., 2007; Baum et al., 2007; Cook et al., 2017; Moore et al., 2011; Rixen et al., 2008).

However, our understanding of the fate of tDOC in rivers, estuaries, and in the ocean, is still limited. Most tDOC is derived from soils, from which it is leached by rainwater, and it is thus rich in lignin and humic substances. Classically, these high-

molecular weight, highly aromatic molecules have been assumed to be inherently refractory to degradation (Bianchi, 2011), which would imply that they should accumulate in the ocean. However, dissolved organic matter (DOM) in the open ocean does not show clear chemical signatures of terrigenous origin, which indicates that terrigenous DOM (tDOM) must be partly remineralised and chemically altered before reaching the open ocean (Bianchi, 2011; Cai, 2011). Although it is now established that tDOM is indeed partly labile to both photo-oxidation (Helms et al., 2014; Miller and Zepp, 1995; Moran et

al., 2000; Spencer et al., 2009; White et al., 2010) and to microbial degradation (Fasching et al., 2014; Leff and Meyer, 1991; Moran and Hodson, 1990; Obernosterer and Benner, 2004; Stutter and Cains, 2016; Ward et al., 2013), we are still far from having a quantitative understanding of tDOM processing in estuaries and shelf seas globally. For example, some studies have reported major losses of tDOM, with 40–50% of the tDOC flux being remineralised on the Louisiana Shelf and in the Eurasian Arctic shelf sea (Fichot and Benner, 2014; Kaiser et al., 2017). Yet in contrast, recent analysis of carbon isotopes in

different molecular weight fractions in the open ocean suggests that a larger proportion of oceanic DOM may have a terrigenous origin than currently thought (Zigah et al., 2017). High-resolution mass spectrometry has also identified new terrigenous biomarkers and shown that they are widely distributed throughout the oceans (Medeiros et al., 2016). Moreover, experimental work has clearly shown that some tDOM fractions are resistant to photo-degradation (Stubbins et al., 2017). Clearly, more work is needed to trace tDOM fluxes through estuaries, and to determine where, how, and to what degree

tDOM is biogeochemically processed.

Because tDOM is rich in chromophoric dissolved organic matter (CDOM), optical measurements are commonly used as proxies to trace tDOM fluxes into the ocean (Chen et al., 2015; Fichot and Benner, 2012; Fichot et al., 2016; Kowalczuk et al., 2003; Osburn et al., 2016; Yamashita et al., 2011). The last decade in particular has seen significant advances in our understanding of how CDOM spectral characteristics vary between tDOM and marine DOM, and how they are affected by

different biogeochemical processes (Hansen et al., 2016; Helms et al., 2008; Helms et al., 2013; Helms et al., 2014; Shank et al., 2005). As a result, CDOM spectral slope coefficients in the ultraviolet (UV) part of the spectrum have emerged as a robust way to trace tDOM fluxes across salinity gradients and to infer biogeochemical transformations of tDOM (Chen et al., 2015; Fichot et al., 2014; Fichot et al., 2016; Helms et al., 2008; Osburn et al., 2016; Stedmon and Markager, 2003).



So far, however, most studies of tDOM fluxes to the sea have focused on North America (Chen et al., 2015; Durako et al., 2010; Fichot and Benner, 2014; Fichot et al., 2014; Leech et al., 2016; Medeiros et al., 2017), Europe (Painter et al., 2018; Rathgeb et al., 2017; Stedmon et al., 2000; Yamashita et al., 2011), and the Arctic (Benner et al., 2005; Dittmar, 2004; Kaiser et al., 2017; Semiletov et al., 2016). Much less work has been conducted in South-East Asia, despite the

disproportionately large fluxes of tDOC through South-East Asia's peatland-draining rivers. Most research in South-East Asia has focused on the peatlands themselves to quantify their extent, carbon stocks, and biogeochemistry (Cobb et al., 2017; Dommain et al., 2014; Gandois et al., 2013; Gandois et al., 2014; Gastaldo, 2010; Page et al., 2011), or has examined just rivers and estuaries, but not traced tDOM further beyond the coast (Alkhatib et al., 2007; Baum et al., 2007; Cook et al., 2017; Harun et al., 2016; Müller et al., 2015; Rixen et al., 2008; Wit et al., 2015). Moreover, most studies of rivers focused

either on the total DOC concentration (Alkhatib et al., 2007; Baum et al., 2007; Cook et al., 2017; Rixen et al., 2008) or on water–air $CO_2$ fluxes (Müller et al., 2015; Müller et al., 2016; Wit et al., 2015). These studies have shown clearly that peatland-draining rivers in Sumatra and Borneo have amongst the highest-reported DOC concentrations from any rivers globally (up to 3,000–5,500 µmol $L^{-1}$, or 36–66 mg C $L^{-1}$). Yet surprisingly, the $CO_2$ fluxes out of these rivers were found to be quite low relative to the extremely high DOC concentrations, implying that most of the tDOC they carry is delivered to

the sea (Wit et al., 2015). To understand the biogeochemical processing of South-East Asian tDOC, more work clearly needs to be done in coastal waters. This need is particularly urgent because most peatlands in South-East Asia have been converted to agricultural use over the past two decades (Miettinen et al., 2016), and such conversion appears to enhance riverine tDOC fluxes by destabilising the peatland C pool (Moore et al., 2013).

Here, we present what is to our knowledge the first analysis of DOC concentrations and CDOM spectral properties in six

rivers and the surrounding coastal sea in the western part of Sarawak, Malaysian Borneo. Samples were collected at three different times of the year to constrain seasonal variability, and photo-degradation experiments were conducted to determine tDOM photo-lability.

## 2 Materials and Methods

### 2.1 Study region and sample collection

Three field expeditions to Sarawak were undertaken in 2017, in early March, in June, and in September. A total of six rivers were sampled in March and in September: these rivers were the Rajang, the Maludam, the Sebuyau, the Simunjan, the Sematan, and the Samunsam (Fig. 1). The June expedition sampled only the Rajang river. On all expeditions, the river estuaries and open coastal waters were sampled (Fig. 1). In September, one sample was also taken in the estuary of a seventh

river, the Lundu river. All station locations and sampling dates, as well as the measured data, are shown in Supplementary Table 1. Four of the rivers (Maludam, Simunjan, Sebuyau, and Samunsam) are blackwater rivers that drain peatlands, while the Sematan and Lundu rivers drain catchments with a higher proportion of mineral soils. The Rajang river drains mineral





soils in its upper reaches until the town of Sibu, from where it branches out into multiple distributary channels (Fig. 1). The distributary channels each have unique names; the main ones (Rajang, Serendeng, and Igan) are identified in Fig. 1. These distributaries are surrounded by extensive peatlands that drain directly into the distributary channels (Staub et al., 2000). All river samples are distinguished below by river name, while marine samples are distinguished by whether they were collected

in the region east of Kuching ("Eastern Region", influenced strongly by the Rajang river outflow), or in the region west of Kuching ("Western Region", influenced by the Samunsam and Sematan rivers). The Western Region is home to coral reefs surrounding the Talang Islands (Fig. 1). The three sampling periods corresponded to the end of the north-east monsoon (March, end of the wettest season of the year), the south-west monsoon (June, lower precipitation), and shortly before the beginning of the north-east monsoon (September, end of the drier season). Monthly precipitation across Sarawak can vary

several-fold across the year, but is only rarely less than 100 mm per month (Sa'adi et al., 2017). Precipitation records were obtained from the weather stations in Sibu, Maludam, and Sematan. Monthly averages were calculated for the period 1999–2017, omitting the small number of months for Maludam and Sematan for which there were days with missing data (there were no missing data in 2017). Monthly precipitation in 2017 was mostly within 1 standard deviation of the mean for 1999–2017 (Fig. 1e), with unusually high and low precipitation only in a few months in Sibu and in Maludam. However, because

precipitation in this region is strongly driven by small-scale convective features, the Sibu precipitation record does not represent the entire Rajang river catchment; rather, we take the precipitation data as showing that 2017 was overall not an unusual year in terms of precipitation. Water temperatures in Sarawak show essentially no seasonal variation (average water temperatures during all expeditions fell within 28.5–29.5° C).

To collect samples in the Rajang river and the Eastern Region, a liveaboard fishing boat was chartered for 4–7 day cruises,

and all samples were filtered and preserved directly on board the boat. All other stations were sampled from small outboard-powered boats, in which case sample waters were stored dark at ambient temperature in insulated boxes on board, and filtered back on land each afternoon/evening. All samples were collected from surface waters within the upper 1 m using either a bucket or a hand-held jug; sampling devices were rinsed thoroughly with sample water at the start of each station. Depending on the parameter to be measured, sample water was decanted into either amber borosilicate glass bottles (DOC

and CDOM) or into HDPE bottles (chlorophyll and total suspended solids).

DOC and CDOM samples were filtered through 0.2-µm pore-size Anodisc filters (47-mm diameter) using an all-glass filtration system that was rinsed with 1 M HCl and ultrapure deionised water (18.2 MΩ cm$^{-1}$, referred to as "DI water" below) in between each sample. Each Anodisc filter was first rinsed by filtering 100–150 mL of DI water and then 50–100 mL of sample water (depending on particle load and DOC concentration), before a further 100–150 mL of sample water

were filtered and taken as the sample. DOC samples (30 mL) were immediately acidified with 100 µL of either 25% $H_3PO_4$ (March expedition) or 50% $H_2SO_4$ (all other samples) to pH <2.0. CDOM samples (30 mL) were preserved with 150 µL of 10 g L$^{-1}$ NaN$_3$, following the REVAMP protocols (Tilstone et al., 2001). DOC and CDOM samples were stored in amber borosilicate EPA vials with PTFE-lined septa at +4° C until analysis (within 1.5 months of collection), although some



samples from the September expedition froze for 1-2 days due to a refrigerator malfunction in the field. However, freezing did not appear to affect the DOC or CDOM results.

Samples (50–1000 mL) for chlorophyll-*a* analysis were filtered onto pre-ashed (450º C, 4 h) 25-mm diameter Whatman GF/F filters, wrapped in aluminium foil, and immediately frozen in a liquid nitrogen dry shipper. They were stored in the dry
shipper until analysis within 6 months of collection.

Samples for total suspended solids (TSS, 50–1000 mL) were filtered onto pre-ashed (450º C, 4 h), pre-rinsed, and then pre-weighed 25-mm diameter Whatman GF/F filters. Filters were rinsed three times with DI water to remove any dissolved salts, and then stored at -20º C in Petri dishes.

Procedural blanks for all parameters were prepared in the field using DI water.

## 2.2 Chemical analyses

### 2.2.1 Dissolved organic carbon analysis

Dissolved organic carbon was analysed on a Shimadzu TOC-L system with a high-salt combustion tube and catalyst, and calibrated using potassium hydrogen phthalate. Samples were analysed as non-purgeable organic carbon after a 5-min sparge time, with 5–7 injections of 150 µL volume. Carrier gas (80 mL min$^{-1}$) was produced by a zero-air generator. Analytical
blanks were prepared freshly for each run using water from an Elga Purelab Flex 3 system (18.2 MΩ cm$^{-1}$, includes a UV lamp and TOC monitor); these blanks were identical to or lower than certified Low-Carbon Water from the Hansell Laboratory, University of Miami. Instrument performance was monitored throughout each run using Deep-Sea Water reference material from the Hansell Laboratory, University of Miami, certified at 42–45 µmol L$^{-1}$ DOC (Batch 16, Lot 11–16). Our analyses consistently yielded very slightly higher values for the reference water, with a long-term mean ± 1 SD of
47 ± 2.0 µmol L$^{-1}$ (n = 51). Procedural blanks prepared in the field almost all contained <10 µmol L$^{-1}$, except for those prepared in between processing blackwater river samples, which contained 13–27 µmol L$^{-1}$; a correction for these procedural blanks was not applied.

### 2.2.2 Chromophoric dissolved organic matter analysis

Samples for CDOM were warmed to room temperature and their absorbance measured from 230–900 nm against a DI water
reference, using a Thermo Evolution 300 dual-beam spectrophotometer. Samples from March were measured in either a 10-cm or a 1-cm pathlength quartz cuvette, or diluted ten-fold with DI water and measured in a 1-cm quartz cuvette (in the case of blackwaters). Samples from June were all measured undiluted in 10-cm or 1-cm cuvettes, and samples from September were measured undiluted in either 10-cm, 1-cm, or 0.2-cm pathlength cuvettes. Prior to analysis, instrument performance was checked according to Mitchell et al. (2000). Reagent blanks were made in the laboratory with 30 mL DI water and 150
µL 10 g L$^{-1}$ NaN$_3$, and blank spectra were subtracted from all measurements. NaN$_3$ was found to absorb significantly in the





region of 230–265 nm, with absorbances around 26 m$^{-1}$ at 230 nm, 4 m$^{-1}$ at 254 nm, but ≤0.1 m$^{-1}$ at wavelengths ≥275 nm. Blank absorbances at wavelengths ≥275 nm were nearly always <10% of sample absorbances, and mostly around 1% or less. All CDOM spectra were baseline-corrected by subtracting the mean absorption from 700–800 nm (Green and Blough, 1994) and converted to Napierian absorption coefficients following Eq. (1):

$a_\lambda = 2.303 \times \frac{A_\lambda}{l}$ ,                                                                                          (1)

where $a_\lambda$ is the absorption coefficient at wavelength $\lambda$, $A_\lambda$ is the absorbance at wavelength $\lambda$, and $l$ is the pathlength of the cuvette in m. These calculations were performed using the R package hyperSpec (Beleites and Sergo, 2018). All of our raw CDOM spectra (as decadic absorbances) are included in Supplementary Table 1, and representative spectra are shown in Supplementary Fig. 1. CDOM spectral slope coefficients were calculated for the intervals 275–295 nm and 350–400 nm as

the absolute value of the slope of a linear regression of $\log_{10}$-transformed absorption coefficients against wavelength, following Helms et al. (2008). The spectral slope ratio was then calculated as the ratio of the slope at 275–295 nm to the slope at 350–400 nm. Specific UV Absorbance at 254 nm (SUVA$_{254}$) was calculated by dividing the sample absorbance at 254 nm, $A_{254}$ m$^{-1}$, by the DOC concentration in mg L$^{-1}$. Note that absorbances by dissolved inorganic constituents such as bromide, sulfide, nitrate, iodide, and molecular oxygen are negligible at wavelengths greater than 250 nm compared to the

high absorbance by CDOM (Fally et al., 2000; Guenther et al., 2001).

### 2.2.3 Conservative mixing models for DOC and CDOM

Two-endmember mixing models for DOC and CDOM could be calculated for the Rajang river, the Samunsam and Sematan rivers, and in September also for the Maludam river. For the other rivers, there was either insufficient variation in salinity (Maludam and Simunjan), or the salinity was influenced strongly by adjacent rivers that were not sampled (Sebuyau, which

drains into the Lupar river estuary). For the Rajang, Sematan, and Samunsam, appropriate end-member stations were identified for river water and for seawater based on station salinity, DOC concentration, and geographical location. In March, rough seas prevented us from sampling a genuine marine end-member station in the Eastern Region, so the marine end-member station from the June expedition was used instead.

Linear mixing models were calculated from the end-member DOC concentrations and CDOM spectra at salinity intervals of

1.0 from salinity 0 (river water) to the salinity of the marine end-member station (29–32.5, depending on location and season). For CDOM, we calculated the full absorption spectrum for each mixture and then re-calculated the corresponding spectral slopes and SUVA$_{254}$ values, yielding non-linear mixing curves (Stedmon and Markager, 2003).

### 2.2.4 Chlorophyll-*a* and total suspended solids analysis

Chlorophyll samples were extracted in 10 mL 90% acetone at -20º C in the dark for 24 h, and fluorescence measured at

excitation 436 nm / emission 680 nm (both with 5 nm bandpass) on a Horiba Fluoromax 4 spectrofluorometer (Welschmeyer, 1994). The fluorescence signal was normalised to the excitation lamp reference intensity and calibrated



against a chlorophyll-*a* standard from spinach (Sigma-Aldrich, C5753). The limit of detection (3 SD of the blank) was <2 ng chlorophyll per filter.

TSS samples were then dried at 75º C for 24 h, cooled to room temperature in a desiccator, and re-weighed. In March and September, they were then ashed at 450º C for 1 hour to remove organic matter and weighed again to determine inorganic

weight. All weighing was performed on a Mettler-Toledo microbalance with ±1 μg accuracy.

### 2.3 Photo-degradation experiments

Four short-term photo-degradation experiments were conducted in the field (one in June, three in September). For each, 1 L of sample water was filtered as for DOC and CDOM samples (using multiple Anodiscs if necessary), and divided into 150 mL quartz bottles with ground quartz stoppers. Dark bottles were wrapped in aluminium foil and black plastic sheets. One

dark and one light control bottle were filled with DI water and treated the same way as sample bottles to check for any systematic contamination (which was not found). All bottles were secured inside a clear, open plastic food storage container and exposed to natural sunlight on the roof of the boat (Rajang and seawater experiments) or in an open clearing on land (Samunsam experiment; incubations were completely unaffected by shade between at least the hours of 08:00 to 16:00). To moderate the temperature during sun exposure, the container was filled with clear seawater to the same level as the samples

in the bottles.

Experiments were run for 3–6 days. Solar radiation was monitored at varying intervals throughout each day with the reference irradiance sensor of a Trios RAMSES hyperspectral radiometer (318–800 nm at 2-nm resolution). We integrated the measured irradiance from 318–450 nm (*i.e.* the portion of the spectrum with highest CDOM absorption), and then integrated this irradiance over time for each day, using exponential averaging to interpolate across measurement gaps, to

estimate the cumulative irradiance from 318–450 nm, in J m$^{-2}$, that samples were exposed to throughout each experiment. This allowed us to account for differences in light intensity between days and between experiments by plotting the observed changes in DOM against actual irradiance, instead of just as a function of time. However, owing to the large diameter and complex geometry of our quartz bottles, the diurnal change in sunlight angle, and the presence of the plastic container, we could not estimate absorbed light doses reliably enough to calculate apparent quantum yields of DOM photo-degradation.

We therefore used the estimated irradiances to help us understand qualitatively how CDOM and DOC changed during sunlight exposure.

During the first three experiments, the radiometer was installed adjacent to the exposed samples and run throughout the experiments. During the experiment with Samunsam river water, the radiometer was in use on board the sampling boat while the samples were being exposed on land; however, there was only little cloud cover during those days, and this was evenly

distributed across land and sea. The Samunsam photo-degradation experiment was then continued for an additional three days, during which no radiometer measurements could be taken. To estimate the approximate irradiances for these days, the amount of cloud cover on each day was noted, and radiometer measurements were taken from previous days that approximately matched the cloud conditions. The integrated irradiance for the Samunsam experiment is therefore less well




constrained than for the other experiments, but since we are not attempting to quantitatively relate DOC degradation to absorbed photon dose, these uncertainties do not compromise our conclusions about the potential photo-lability of tDOM in Sarawak.

**3 Results**

**3.1 Concentrations of DOC**

Concentrations of DOC differed significantly between rivers, with highest freshwater concentrations (salinity = 0) in the Maludam (3,100–4,400 µmol L$^{-1}$), and lowest concentrations in the Rajang (120–250 µmol L$^{-1}$) (Fig. 2a–e). Seasonal differences were clearly apparent in the river samples, but did not show a consistent direction: the Sematan and Samunsam

rivers carried ≥50% higher DOC concentrations in September than in March, the Sebuyau river had marginally higher concentrations in September than in March, while the Maludam and Simunjan rivers had 20%–40% lower DOC concentrations in September than in March. Seasonality in the Rajang river is not apparent in Fig. 2a, possibly because of variability between distributary channels.

In March, DOC concentrations at freshwater stations in the Maludam and Sebuyau increased with distance downstream, but

decreased slightly in the Simunjan (Fig. 3a–c). In September, DOC concentrations varied little with distance downstream in these three blackwater rivers (Fig. 3a–c). In the Rajang river, DOC concentrations at salinity 0 in each individual channel were somewhat higher in March than in September (Fig. 3g–i; June data are not shown because only one station had salinity 0 in June). Notably, DOC concentrations in the Rajang delta increased substantially with distance downstream in each of the distributary channels (Fig. 3g–i), with concentrations doubling to around 240 µmol L$^{-1}$ during passage through the northern-

most distributary, the Igan.

DOC concentrations decreased with increasing salinity in all river estuaries (Fig. 2a–e). In the Samunsam, Sematan, and Maludam rivers this decrease closely followed the predictions from a two-end-member mixing model, with the single sample from the Lundu river plotting very close to the Sematan mixing model. For the Sebuyau river, a conservative mixing model could not be constructed, because it discharges into the estuary of the larger Lupar river (Fig. 1), for which the freshwater

end-member DOC concentration was not measured. For the Simunjan river, and for the Maludam river in March, too few data from brackish waters were available to construct a reliable mixing model.

In contrast, the DOC concentrations in the Rajang delta do not fit the conservative mixing models well: most of the March and June data from brackish stations have higher DOC concentrations than expected, and in fact are closer to predictions from the September mixing model (Fig. 2a). The September mixing model was calculated using the DOC concentration of

the two northern-most stations in the Igan distributary as the freshwater end-member, which had the highest DOC concentrations of any of the Rajang river stations.





Some seasonality in DOC concentration was seen at the stations furthest off-shore, which had the highest salinities (29.0–32.5). This was most evident in the Western Region: in March, the stations with highest salinities (28.9 and 29.0) contained 93 and 87 µmol L$^{-1}$ DOC, while in September the same stations had 76 and 78 µmol L$^{-1}$ DOC, and salinities >32.0. In the Eastern Region, the highest off-shore salinities in June were 30.2–32.1 with DOC concentrations of 81–99 µmol L$^{-1}$, while in

September the DOC concentrations were lower at 76–83 µmol L$^{-1}$ with salinities of 29.7–31.6 (except for one station with 88 µmol L$^{-1}$ DOC).

### 3.2 Spectral characteristics of CDOM

#### 3.2.1 CDOM absorption coefficient

CDOM concentrations, quantified as $a_{350}$, were high throughout our study region. Nearly all samples in the blackwater rivers
had $a_{350}$ values >50 m$^{-1}$, with samples from the Maludam reaching 210 m$^{-1}$ (Fig. 2f–j). Lower values were found in the Rajang and Sematan rivers, between 3–11 m$^{-1}$. The lowest $a_{350}$ value (0.23 m$^{-1}$) was found in the furthest off-shore station in the Western region in September. The mixing behaviour of $a_{350}$ closely mirrored that of DOC in each of the rivers (Fig. 2).

#### 3.2.1 Spectral slopes

The CDOM spectral slope from 275–295 nm ($S_{275–295}$) was low in all freshwater samples, ranging from 0.0102–0.0144, and
increased with salinity to a maximum of 0.0254 (Fig. 4a–e). $S_{275–295}$ was somewhat lower in most of the blackwater samples (Samunsam, Maludam, Sebuyau, and Simunjan) than in the Rajang and Sematan. Seasonal differences were clearly seen in the marine samples in the Western Region, with $S_{275–295}$ in March always below 0.0200, but up to 0.0254 in September (Fig. 4b). In the Rajang distributaries, $S_{275-295}$ was lower in March than in September (Fig. 3j–l), but no clear seasonality was seen in the Eastern Region marine samples (Fig. 4a), although we were unable to collect many marine samples in March due to
rough seas.

In the Samunsam and Maludam rivers, $S_{275–295}$ closely followed the conservative mixing models, but this was not the case in the Rajang and the Sematan rivers (Fig. 4a–c). In the Rajang, samples at salinities 3–20 typically had higher $S_{275–295}$ than predicted by the mixing models (except in June), while many samples at salinities >20 had $S_{275–295}$ values that were lower than predicted by the mixing models. Similarly, in the Sematan river in March, samples in brackish water up to salinity 20
showed higher $S_{275–295}$ than expected from conservative mixing. In September, we were unable to sample fully freshwater in the Sematan river owing to the timing of the tides, so the freshest sample still had a salinity of 3.4, and may therefore have already been affected by any non-conservative processes in the estuary. However, all samples in the Western Region with salinities >25 fell very closely between the conservative mixing lines for the Sematan and Samunsam rivers.

The spectral slope from 350–400 nm ($S_{350–400}$) showed more complex trends: freshwater samples had values mostly between
0.014 and 0.018, while brackish and fully marine waters spanned a greater range of 0.0076–0.0206 (Fig. 4f–j). The marine end-member stations in the Eastern and Western Regions both had lower $S_{350–400}$ than the river end-members in September,



but had higher values (Western Region) or nearly identical values (Eastern Region) in March and June. In the Rajang and Sematan rivers, $S_{350-400}$ showed conservative mixing up to salinities of 20–25, but was lower than predicted by conservative mixing in the blackwater Samunsam and Maludam rivers (Fig. 4f–h). At salinities >20–25, many samples clearly depart from the conservative mixing models, except for samples in March in both regions.

However, the spectral slope ratio, $S_R$, showed trends very similar to $S_{275-295}$, *i.e.* low values in river waters (0.601–0.867) and higher values in marine waters with salinity >25 (0.786–2.33, Fig. 4k–o). In brackish waters, $S_R$ was typically slightly greater than predicted by the conservative mixing models, especially in March in the Rajang, Sematan, and Samunsam rivers, and in September in the Rajang, Samunsam, and Maludam rivers (Fig. 4k–m).

### 3.2.2 Specific UV absorbance

The specific UV absorbance at 254 nm ($SUVA_{254}$) was higher in river samples (3.08–6.89 at salinity 0) than in marine samples (0.81–5.00 at salinity >25), and decreased with salinity for all rivers and seasons (Fig. 4p–t). $SUVA_{254}$ was somewhat higher in the Rajang and in the Simunjan in March than in June or September, but otherwise seasonal differences in the rivers were not pronounced. However, as for the other CDOM parameters, there was a clear difference between March and September in the marine samples from the Western Region (Fig. 4q). The data from the Maludam, Sematan, and

Samunsam rivers closely followed the conservative mixing lines, while in the Rajang, the majority of brackish samples with salinity >20 showed somewhat greater $SUVA_{254}$ than expected from conservative mixing (Fig. 4p–r). Because sodium azide contributes a relatively high blank absorbance at 254 nm but not beyond 270 nm, we compared our $SUVA_{254}$ values to the specific UV absorbance at 280 nm, $SUVA_{280}$, for each sample. We found a very strong, linear relationship between $SUVA_{280}$ and $SUVA_{254}$ for the entire dataset across rivers and seasons, with $SUVA_{280} = 0.7926*SUVA_{254} - 0.0155$ ($r^2 = 0.990$, p

<0.001, n = 156), suggesting that our $SUVA_{254}$ measurements were not compromised by the $NaN_3$ blank.

### 3.2.3 Relationships between DOC and CDOM

The CDOM concentration, as $a_{350}$, was closely related to the DOC concentration for the entire dataset (Fig. 5a). $S_{275-295}$ was also strongly related to DOC, though with somewhat greater scatter at DOC concentrations greater than about 150 µmol L$^{-1}$ (Fig. 5b). Consequently, there was also a strong relationship between $S_{275-295}$ and $a_{350}$, although also with more scatter

wherever $a_{350}$ >10 m$^{-1}$ (Fig. 5c). There was no seasonal variation in any of these relationships.

### 3.3 Photo-degradation of DOC and CDOM

DOM from the Rajang and Samunsam rivers was photo-labile, with DOC and CDOM decreasing after sunlight exposure. In contrast, marine water collected in the Eastern Region only showed some changes in the CDOM spectrum, but no decrease in DOC (Fig. 6, Table 1). Daily irradiances, integrated from 318–450 nm, ranged from 0.92 to 3.00 MJ m$^{-2}$, with cumulative

irradiances for each experiment ranging from 5–11 MJ m$^{-2}$. Irradiance data for each day are shown in Supplementary Fig. 2. In practice, plotting our data against estimated cumulative irradiance showed the same trends as plotting simply against time



of exposure (Supplementary Fig. 3), although we estimate that the Samunsam water received a slightly higher irradiance over five days than the marine water over six days, and that the two Rajang experiments differed by about 20% in irradiance despite both lasting three days.

The Rajang water in June lost $16.1 \pm 0.5$ µmol L$^{-1}$ DOC by the end of the experiment (mean $\pm$ 1 SD, representing 8.8%–
9.4% of the starting DOC). $S_{275–295}$ and $S_R$ both increased, while $S_{350–400}$ remained essentially unchanged, and SUVA$_{254}$ decreased slightly (Fig. 6). In September, we found very similar changes in the Rajang water after sunlight exposure: $18.9 \pm 6.1$ µmol L$^{-1}$ DOC were lost (mean $\pm$ 1 SD, representing 5.6%–10.7% of starting DOC), and $S_{275–295}$ and $S_R$ increased by amounts similar to June. Although $S_{350–400}$ decreased slightly relative to the initial sample, there was no difference between light and dark bottles in this parameter. SUVA$_{254}$ decreased slightly in the light bottles, and actually increased somewhat in
the dark bottles. Marine water showed no change in DOC upon light exposure, although light bottles had very slightly higher DOC concentrations than dark bottles at the end of the experiment (by 4.5–6.1 µmol L$^{-1}$). However, $S_{275–295}$ increased strongly due to light exposure, reaching values higher than seen in any of our environmental samples (>0.030). $S_{350–400}$ increased both in light and dark bottles relative to the initial sample, and $S_R$ consequently dropped in the dark bottles but remained relatively constant in the light bottles. SUVA$_{254}$ decreased slightly after light exposure. The greatest effects of
photo-degradation were seen in the Samunsam river blackwater, with a decrease in DOC by $432 \pm 42$ µmol L$^{-1}$ (mean $\pm$ 1 SD, representing 21%–26% of initial DOC). $S_{275–295}$ and $S_R$ both increased, $S_{350–400}$ decreased, but SUVA$_{254}$ remained essentially unchanged (Fig. 6). Notably, DOC showed a linear decrease with cumulative irradiance in all three river water experiments, suggesting that more DOC could have been mineralised if sunlight exposure had continued.

### 3.4 Distributions of chlorophyll-*a* and suspended sediments

Chlorophyll-*a* concentrations were mostly <3 µg L$^{-1}$ throughout the region, and never exceeded 5.5 µg L$^{-1}$, indicating quite oligotrophic conditions (Fig. 7a–e). Concentrations in the rivers at salinity 0 were always <1 µg L$^{-1}$ except in the Simunjan (up to 3.8 µg L$^{-1}$), and higher values were generally found in the estuaries at salinities between 10 and 25.

Total suspended solids in the Rajang reached values up to nearly 400 mg L$^{-1}$, with values in the brackish waters of the Rajang delta varying mostly between 10–70 mg L$^{-1}$ (Fig. 7f–i). More than 90% by weight of this material was inorganic. The
other rivers, and the most distant marine samples, all contained far lower TSS concentrations, but the estuaries always had >10 mg L$^{-1}$.

### 4 Discussion

### 4.1 Distribution of DOM within and between rivers

Rivers in Sarawak clearly differ substantially in their DOM concentrations. All of the blackwater rivers had DOC concentrations above 1,200 µmol L$^{-1}$, with highest values in the Maludam river. These results are consistent with previous



measurements in the Maludam (Müller et al., 2015), and in other blackwater rivers in South-East Asia (Alkhatib et al., 2007; Cook et al., 2017; Harun et al., 2016; Moore et al., 2011; Rixen et al., 2008; Wit et al., 2015), but they are high compared to DOC measurements in blackwaters from other continents, which are typically below 2,000 μmol L$^{-1}$ (Lawrenz et al., 2010; Leech et al., 2016). The Maludam, Sebuyau, and Simunjan river drain peatlands along most of their catchments (Müller et

al., 2016), while the Samunsam river drains an extensive area of peatland in its upper reaches. The lower DOC concentrations in the Rajang, Sematan, and Lundu rivers are closer to concentrations reported from the Lupar and Saribas (mostly <500 μmol L$^{-1}$), the two larger rivers that flank the Maludam peat dome (Fig. 1) (Müller et al., 2016). The Rajang river drains mineral soils along most of its catchment, and peatlands (up to several metres thick) are only found in the delta surrounding the distributaries (Gastaldo, 2010; Staub et al., 2000). The Sematan and Lundu river catchments also have at

most limited peat deposits. The pronounced increase in DOC concentration with distance downstream in the three main Rajang distributary channels clearly shows that there are large organic matter inputs into the distributaries. Given the very low chlorophyll-*a* concentrations in the Rajang delta, we can rule out a major autochthonous source of DOC. Bacterial solubilisation of particulate organic carbon (POC) is a possible *in-situ* source of DOC, but our CDOM data did not indicate a substantial bacterial DOM source (see Section 4.4). Photochemical POC solubilisation could also produce DOC *in situ*

(Kieber et al., 2006; Mayer et al., 2006). However, estimated rates of photochemical POC solubilisation under realistic conditions of light penetration are only around 4–6 mmol m$^{-2}$ d$^{-1}$ (Kieber et al., 2006; Riggsbee et al., 2008), which is too low to explain the DOC increase we observed, given the likely short transit time of water through the Rajang delta (see Section 4.3.2). Instead, the DOC input most likely originates from the peatlands in the delta. Peatlands are found throughout the delta, but are most extensive and deep along the Igan distributary (Gastaldo, 2010; Staub et al., 2000), and the Igan also

showed the greatest increase in DOC with distance downstream of all the Rajang distributaries, consistent with our hypothesis of a peatland DOM source to the Rajang delta. However, future work should explicitly address the possibility of POM solubilisation to DOM in South-East Asian rivers.

None of the river catchments consist of genuinely pristine peat swamps: much of the peatland surrounding the Simunjan, Sebuyau, and Rajang rivers has been converted to oil palm plantations, and even the less impacted Maludam and Samunsam

catchments have been disturbed by logging. Anthropogenic disturbance has been shown to increase the loss of DOC from peatlands at field sites in central Borneo (Moore et al., 2013), and Harun et al. (2016) noted differences in DOM quality between agricultural and natural peatland sites in eastern Borneo. However, our study was not designed specifically to address the question of whether anthropogenic disturbance affects DOM concentrations or quality in the rivers; further work would be needed to determine impacts of land-use changes.

Our CDOM data show that all rivers were characterised by high levels of tDOM, with the blackwater rivers in particular having extremely high absorption coefficients, very low S$_{275-295}$ values, and and high SUVA$_{254}$. S$_{275-295}$ is now well established as an optical tracer for tDOC in estuarine and marine waters (Fichot and Benner, 2011, 2012; Helms et al., 2008), and is inversely related to the mean molecular weight of DOM in a sample (Helms et al., 2008). SUVA$_{254}$ is positively related to the aromaticity of DOM (Traina et al., 1990; Weishaar et al., 2003). Our values for S$_{275-295}$ and SUVA$_{254}$ in the

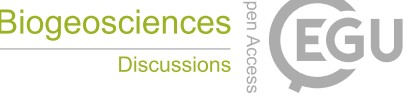



rivers are, respectively, on the low and the high end of values reported from other freshwaters, especially in our blackwater rivers (Fichot and Benner, 2014; Helms et al., 2008; Leech et al., 2016; Massicotte et al., 2017). These river systems in Sarawak are therefore characterised by DOM with high average molecular weight and high aromaticity, consistent with a terrigenous rather than aquatic source. Moreover, even the clearest marine waters that we sampled had $S_{275-295}$ no greater

than about 0.025, which is consistent with a some of the DOM in these waters having a terrigenous origin.

Riverine DOC concentrations do not always show a simple relationship with precipitation and river discharge, especially in peatland-draining rivers (Clark et al., 2007). Previous DOC measurements in the Saribas and Lupar rivers in Sarawak did not show a consistent seasonality (Müller et al., 2016). Our data indicate that DOC concentrations at the end of the wettest season (March) were higher than at the end of the drier season (September) in the Rajang, Maludam, and Simunjan rivers,

but by different amounts. In contrast, the Samunsam, Sematan, and Sebuyau showed lower concentrations in March than in September, but again by different amounts between the rivers. These differences probably reflect differences in catchment hydrology, given that precipitation in 2017 was mostly close to the 18-year mean. However, it is possible that the high precipitation in Maludam in September contributed to lowering the September DOC concentrations, because all of the excess precipitation (relative to the mean) fell over two consecutive days, and less than one week before we sampled. Because the

Maludam is a very small river, the precipitation record from Maludam village is likely to reflect rainfall across the Maludam catchment. This is unlikely to be the case for the Rajang river; hence, it is questionable whether the elevated February precipitation in Sibu would have affected our March data for the Rajang.

Overall, our data clearly show that there is inconsistent seasonality in riverine DOC concentrations across Sarawak, while precipitation is relatively high year-round. Sarawak is thus clearly characterised by high tDOC fluxes to sea in all seasons, as

reflected by the low $S_{275-295}$ values at all of our marine stations. Nevertheless, we found a stronger terrigenous signal during the north-east monsoon in the Western region marine samples, although we cannot say whether this reflects a seasonal change in the magnitude of the tDOC flux to sea, or perhaps a seasonal difference in the degradation rate of tDOC at sea. For example, rougher, more turbid seas and greater cloud cover during the north-east monsoon might reduce the solar irradiance underwater and thus reduce the rate of photo-degradation.

Despite the fact that there are large differences in CDOM concentration and some differences in CDOM spectral properties between rivers, we found very strong, overall relationships between DOC concentration, $a_{350}$, and $S_{275-295}$ in our dataset. Unlike, for example, in the northern Gulf of Mexico (Fichot and Benner, 2011), there was no seasonal variability in these relationships. This suggests that there are no strong seasonal changes in tDOM composition within our study region, despite the seasonal concentration differences discussed above.

**4.2 Photo-lability of riverine DOM**

Our experiments clearly showed that DOM in Sarawak rivers is photo-labile, with both DOC mineralisation and substantial changes to the CDOM absorption spectrum ocurring within days of sunlight exposure. The linearity of DOC loss with cumulative irradiation in our river samples suggests that our experiments were too short for all photo-labile DOC to be lost;





our results are therefore an under-estimate of the total proportion of tDOC in Sarawak rivers that can potentially be photo-mineralised (of course, it must be noted that the *rate* of tDOC photo-oxidation in our incubations was likely much higher than the true *in-situ* rate of photo-oxidation, at least in the highly light absorbant rivers). Only the marine sample did not lose DOC upon exposure to sunlight, although this sample did show a significant increase in $S_{275-295}$ to higher values than found

at any of our stations. This increase in spectral slope for marine CDOM further suggests that there was a terrigenous fraction of DOM even at those stations farthest off-shore.

High photo-lability of tDOM has been shown in many cases, with freshwater samples from North America (Gao and Zepp, 1998; Helms et al., 2008; Helms et al., 2014; Miller and Zepp, 1995; White et al., 2010), Africa (Spencer et al., 2009), and the Arctic (Stubbins et al., 2017) showing loss of DOC and changes in absorption spectra upon solar irradiation. In some

cases, however, riverine tDOC was found to be resistant to photo-degradation, possibly because the photo-labile fraction had already degraded upstream of the sampling site (Chupakova et al., 2018). Phytoplankton-produced DOC and marine surface-water DOC are typically not photo-mineralised (Obernosterer and Benner, 2004; Ziegler and Benner, 2000), although DOC from aphotic deep-sea samples has been shown to be photo-labile (Helms et al., 2013). The changes that we observed in all our samples in $S_{275-295}$ (pronounced increase) and in $S_{350-400}$ (decrease or no change relative to dark bottles) upon sunlight

exposure, and the resulting increases in $S_R$, are also consistent with photo-degradation studies of tDOM in other regions (Helms et al., 2013; Spencer et al., 2009). Our data thus validate these optical measures as indicators of photo-degradation also in South-East Asia. Interestingly, the Samunsam blackwater sample did not show a change in $SUVA_{254}$ upon solar irradiation. Given the very high $SUVA_{254}$ of this sample, it is possible that the contribution of aromatic molecules to the total DOC was so high that the degradation of aromatic moieties was proportional to the overall loss of DOC during photo-

exposure.

Our experimental results thus indicate that tDOM in rivers in Sarawak contains a significant photo-labile fraction that is partly photo-modified and partly photo-mineralised. Because South-East Asian peatlands are predominantly found in coastal lowlands (Dommain et al., 2014), peatland-derived DOM probably has too short a residence time in rivers for significant photo-degradation to occur in the rivers before it reaches the sea, unlike in some Arctic rivers (Chupakova et al., 2018). We

therefore suggest that most photo-chemical transformations of tDOC in Sarawak likely take place after tDOC reaches the sea, rather than inside the rivers and estuaries.

### 4.3 Mixing of riverine DOM with marine water

### 4.3.1 Conservative mixing

In the Maludam, Samunsam, and Sematan estuaries, DOC and most CDOM parameters showed conservative mixing

between river water and seawater. Conservative behaviour of tDOM is often reported from estuaries elsewhere (Chen et al., 2015; Kowalczuk et al., 2003; Rochelle-Newall and Fisher, 2002; Yamashita et al., 2011), including the few South-East Asian rivers that have been studied to date (Alkhatib et al., 2007; Baum et al., 2007; Rixen et al., 2008). Even though tDOM



is now recognised as being less refractory than previously assumed (Bauer et al., 2013; Bianchi, 2011; Cai, 2011), as also shown by our photo-degradation experiments, the removal time-scales of tDOM by photochemical, biological, and other processes are clearly longer than the transit times through these river systems.

The marine waters in the Western Region showed a clear seasonal difference in CDOM spectral characteristics and salinity.

The only significant rivers in this region are the Samunsam, Sematan, and Lundu. In spite of their small size, these rivers clearly deliver enough tDOM to measurably impact the optical characteristics of the DOM pool in coastal waters 10–20 km beyond the river mouths, including on the coral reefs surrounding the Talang Islands. Even our highest values of $S_{275\text{-}295}$ are actually on the low end of values reported from marine samples (usually 0.020–0.050, Stedmon and Nelson (2015)), suggesting that tDOM contributes significantly to the total DOM pool in coastal waters in Sarawak. Beyond the immediate

river plumes, these coastal waters have low concentrations of suspended sediments and of chlorophyll, which means that tDOM delivery might act as an important control over the underwater light availability in this region, and thus over coastal productivity (Cherukuru et al., 2014; Durako et al., 2010; Lawrenz et al., 2010).

**4.3.2 Possible non-conservative mixing in the Rajang**

In the Rajang delta, we found evidence of non-conservative mixing, with most of the brackish waters having higher DOC

concentrations than expected. We attribute this to continued input of tDOC from the extensive peatlands surrounding all of the distributaries (Gastaldo, 2010; Staub et al., 2000), since we observed increasing DOC concentration with distance downstream in each distributary. Moreover, several of the Eastern Region marine samples, with salinities >30, had higher DOC concentrations than predicted by the mixing models, possibly indicating that DOC-rich run-off from peatlands also flows directly into the coastal sea (*e.g.* from peatlands on Pulau Bruit, Staub et al. (2000)). Higher DOC concentrations than

predicted by conservative mixing have sometimes been noted, for example, in the Chesapeake Bay, and attributed to *in situ* production of DOC by phytoplankton (Rochelle-Newall and Fisher, 2002). However, chlorophyll-*a* concentrations in our study region very rarely exceeded 2.5 μg $L^{-1}$, and were always below 1 μg $L^{-1}$ in the Rajang river, which rules out phytoplankton production as a major source of riverine DOC.

Given this large input of likely peat-derived DOC into the Rajang Delta, one would also expect non-conservative behaviour

of CDOM. Surprisingly, however, the majority of brackish samples actually had higher $S_{275\text{-}295}$ values than predicted, contrary to what one would expect from the addition of peatland-derived DOM (indeed, all other rivers had lower $S_{275\text{-}295}$ than our Rajang samples). These results suggest that there is selective removal of a high-molecular weight CDOM fraction within the Rajang delta, despite the continued input of tDOM in the distributaries. Our data from the Sematan river in March actually hint at a similar increase in $S_{275–295}$ than expected from conservative mixing, although our dataset from this river is

too limited to conclude this with confidence.

Non-conservative removal of tDOM in estuaries can occur due to photo-degradation and microbial degradation, flocculation due to rising salinity, or adsorption to sediments. We hypothesise that adsorption of tDOM to suspended inorganic sediments is the most likely explanation for the non-conservative increase in $S_{275–295}$. We measured extremely high suspended inorganic





sediment in all of the Rajang distributaries on all expeditions (100–360 mg L$^{-1}$), which is consistent with previous data (Staub et al., 2000). Such concentrations of sediments have been shown to lower CDOM absorption coefficients and to increase the CDOM spectral slope of estuarine tDOM samples in laboratory incubations (Shank et al., 2005; Uher et al., 2001). While flocculation of tDOM due to rising salinity can occur (Sholkovitz et al., 1978; Uher et al., 2001), we would

also expect this process to affect all of the other, less sediment-rich, rivers similarly, but this was not the case (except, possibly, in the Sematan estuary). Photo-degradation is unlikely to account for CDOM removal in the Rajang delta, because the high sediment loads attenuate light very strongly; Secchi depths, measured on two of our expeditions in the Rajang, were typically in the range of 10–30 cm. Although bacterial degradation of tDOM almost certainly does take place in the Rajang, the transit time of river water through the distributaries is probably too short for biological degradation to account for the

observed changes: the total distance through the distributaries to the coast is around 80–120 km, and current speeds on the order of 1-2 knots were typical, implying a total transit time to sea of just a few days. Sediment adsorption, however, has been shown to alter CDOM spectra within hours to days (Shank et al., 2005; Uher et al., 2001).

### 4.4 Biogeochemical processing of tDOM

Both photo-degradation and microbial degradation typically cause increases in $S_{275-295}$ (Helms et al., 2014; Lu et al., 2016;

Spencer et al., 2009). However, photo-degradation concomitantly reduces $S_{350-400}$, and thus increases $S_R$ (Helms et al., 2013; Helms et al., 2014; Spencer et al., 2009), as seen in all of our photodegradation experiments. In contrast, incubation experiments suggest that bacterial processing of tDOM either has limited impact on, or actually increases, $S_{350-400}$ (Hansen et al., 2016; Lu et al., 2016). Similarly, Moran et al. (2000) found that CDOM spectral slopes over a larger wavelength range are increased by photo-degradation, but reduced by microbial degradation. The fact that $S_R$ showed a very clear increase with

salinity in all rivers, with marine samples mostly showing values >1.0, suggests that photo-degradation plays an important role in processing tDOM in this region. However, the erratic variation seen in $S_{350-400}$, with marine samples having either higher or lower values than river water, may be indicative of a role for microbial tDOM degradation as well. Typically, photo-degradation makes tDOM more labile to microbial remineralisation (Kieber et al., 1989; Miller and Moran, 1997; Obernosterer and Benner, 2004), which makes it hard to disentangle the importance of the two processes based only on

CDOM measurements. While our CDOM data are consistent with an important role for photo-degradation, estimating the amount of tDOC that is actually photo-oxidised to inorganic carbon would require estimates of the water residence time, solar irradiation, and light attenuation coefficients, which are not available for Sarawak at the present time. Based on such a calculation, Fichot and Benner (2014) estimated that the majority of tDOC delivered by the Mississippi river to the Gulf of Mexico is actually remineralised by bacteria, even though the tDOC in their study region was found to be photo-labile in

irradiation experiments.

Importantly, our data suggest that most of the tDOM that is lost from South-East Asia's coastal peatlands is transferred to the seas on the Sunda Shelf, where it becomes part of the global ocean circulation *via* the different branches of the Indonesian Throughflow. However, owing to the large geographical extent of the Sunda Shelf, the residence time of peatland tDOM on



the shelf is likely to be at least several months to a year. During this time, the tDOM always experiences temperatures of 25–30º C (promoting rapid microbial metabolism), and is also likely exposed to relatively high doses of solar irradiation, because these oligotrophic, tropical waters are relatively clear and shallow (mostly <100 m deep). The geographical extent and oceanography of the Sunda Shelf seas are therefore likely to strongly promote the remineralisation of tDOM before this

material enters the open ocean. Remineralisation of tDOC would explain the high $pCO_2$ over-saturation reported by Kartadikaria et al. (2015) across the Java Sea, which is thus consistent with our results.

## 5 Conclusions

We have reported the first seasonally-resolved study of DOC and CDOM for South-East Asia that includes peatland-draining

rivers, estuaries, and coastal waters. Most of the rivers we sampled carried very high concentrations of tDOC and CDOM that showed conservative mixing with seawater in the estuaries. Non-conservative mixing was only found in the Rajang river delta, where our data point towards increasing inputs of tDOM from peatlands along the delta, but also to the removal of a high-molecular weight fraction, probably due to adsorption to sediments. Seasonality in tDOM concentrations differed between rivers, but our CDOM data showed that tDOM concentrations were higher at all marine stations in the western part

of our study region during the north-east monsoon. Overall, our CDOM spectral slope coefficients are consistent with a significant contribution by tDOM to the total DOM pool even at our marine end-member stations, but also suggest that photo-degradation plays an important role in the biogeochemical processing of tDOM in Sarawak's coastal waters. This is also supported by our direct experimental evidence showing that tDOM from rivers in Sarawak is both remineralised and altered upon solar irradiation. Our results therefore suggest that much of the biogeochemical processing of peatland-derived

tDOM in South-East Asia may take place in shelf seas rather than rivers.

*Acknowledgements.* We would like to thank the Sarawak Forestry Department and Sarawak Biodiversity Centre for permission to conduct collaborative research in Sarawak waters under permit numbers NPW.907.4.4(Jld.14)-161, Park

Permit No WL83/2017, and SBC-RA-0097-MM. We are very grateful to the boatmen who helped us to collect samples, in particular Lukas Chin, Captain Juble, and their crew during the Rajang river and Eastern Region cruises, and Minhad and Pak Mat while sampling the Western Region. We are grateful to Claire Evans and Joost Brandsma for their participation in planning the overall research project and helping to lead expeditions to the Maludam, Sebuyau, and Simunjan rivers. Faddrine Jang, Edwin Sia, Gonzalo Carrasco, Jack Sim, Akhmetzada Kargazhanov, Florina Richard, Faith Chaya, Noor

Iskandar Noor Azhar, and Fakharuddin Muhamad assisted greatly during fieldwork and with logistics. The Sarawak Department of Irrigation and Drainage provided precipitation data. P.M. acknowledges funding through a Nanyang



Technological University Start-Up Grant, and a Tier 1 grant from the Singapore Ministry of Education's Academic Research Fund (RG 175/16).





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





Table 1: Summary of results from photo-degradation experiments. All values are mean ± SD for each treatment. Day 0 data are the values measured at the corresponding station from which water for each experiment was taken. All irradiance data are in Joules m$^{-2}$, integrated from 318–450 nm.

| | | | | | | | |
|---|---|---|---|---|---|---|---|
| **Rajang experiment June** | | | | | | | |
| | Cumulative irradiance | DOC, µmol L$^{-1}$ | | $S_{275-295}$ | | $S_{350-400}$ | |
| Day | | Light | Dark | Light | Dark | Light | Dark |
| 0 | 0 | 178 | | 0.0132 | | 0.0172 | |
| 1 | 1.93*10$^6$ | 171 ± 0.31 | 177 ± 1.34 | 0.0164 ± 6.4*10$^{-5}$ | 0.0131 ± 3.9*10$^{-5}$ | 0.0174 ± 1.5*10$^{-4}$ | 0.0171 ± 1.1*10$^{-4}$ |
| 3 | 6.53*10$^6$ | 162 ± 0.51 | 177 ± 0.07 | 0.0183 ± 1.6*10$^{-4}$ | 0.0132 ± 6.3*10$^{-6}$ | 0.0174 ± 4.4*10$^{-5}$ | 0.0169 ± 6.9*10$^{-5}$ |
| **Rajang experiment September** | | | | | | | |
| | Cumulative irradiance | DOC, µmol L$^{-1}$ | | $S_{275-295}$ | | $S_{350-400}$ | |
| Day | | Light | Dark | Light | Dark | Light | Dark |
| 0 | 0 | 238 | | 0.0130 | | 0.0171 | |
| 1 | 1.10*10$^6$ | 230 ± 1.4 | 244 ± 4.8 | 0.0146 ± 2.0*10$^{-4}$ | 0.0129 ± 1.3*10$^{-4}$ | 0.0152 ± 1.1*10$^{-4}$ | 0.0165 ± 2.2*10$^{-3}$ |
| 3 | 4.72*10$^6$ | 219 ± 6.1 | 240 ± 4.6 | 0.0168 ± 3.7*10$^{-4}$ | 0.0123 ± 5.8*10$^{-4}$ | 0.0156 ± 4.9*10$^{-4}$ | 0.0152 ± 8.1*10$^{-4}$ |
| **Marine experiment September** | | | | | | | |
| | Cumulative irradiance | DOC, µmol L$^{-1}$ | | $S_{275-295}$ | | $S_{350-400}$ | |
| Day | | Light | Dark | Light | Dark | Light | Dark |
| 0 | 0 | 83 | | 0.0228 | | 0.0137 | |
| 4 | 6.08*10$^6$ | 85 ± 0.8 | 79 ± 1.6 | 0.0299 ± 5.2*10$^{-4}$ | 0.0246 ± 9.1*10$^{-4}$ | 0.0178 ± 1.5*10$^{-4}$ | 0.0177 ± 6.8*10$^{-4}$ |
| 6 | 9.71*10$^6$ | 86 ± 1.2 | 82 ± 1.6 | 0.0307 ± 2.5*10$^{-4}$ | 0.0245 ± 2.3*10$^{-4}$ | 0.0177 ± 8.9*10$^{-4}$ | 0.0174 ± 7.1*10$^{-4}$ |
| **Samunsam experiment September** | | | | | | | |
| | Cumulative irradiance | DOC, µmol L$^{-1}$ | | $S_{275-295}$ | | $S_{350-400}$ | |
| Day | | Light | Dark | Light | Dark | Light | Dark |
| 0 | 0 | 1799 | | 0.0109 | | 0.0160 | |
| 1 | 2.99*10$^6$ | 1584 ± 50 | 1753 ± 11 | 0.0122 ± 1.3*10$^{-4}$ | 0.0112 ± 3.2*10$^{-5}$ | 0.0151 ± 4.2*10$^{-5}$ | 0.0163 ± 6.4*10$^{-5}$ |
| 3 | 7.13*10$^6$ | 1472 ± 30 | 1782 ± 38 | 0.0129 ± 7.0*10$^{-5}$ | 0.0109 ± 5.6*10$^{-4}$ | 0.0147 ± 8.1*10$^{-5}$ | 0.0158 ± 6.1*10$^{-4}$ |
| 5 | 11.2*10$^6$ | 1381 ± 42 | 1773 ± 17 | 0.0132 ± 1.2*10$^{-4}$ | 0.0109 ± 7.1*10$^{-5}$ | 0.0143 ± 1.8*10$^{-4}$ | 0.0158 ± 9.1*10$^{-5}$ |





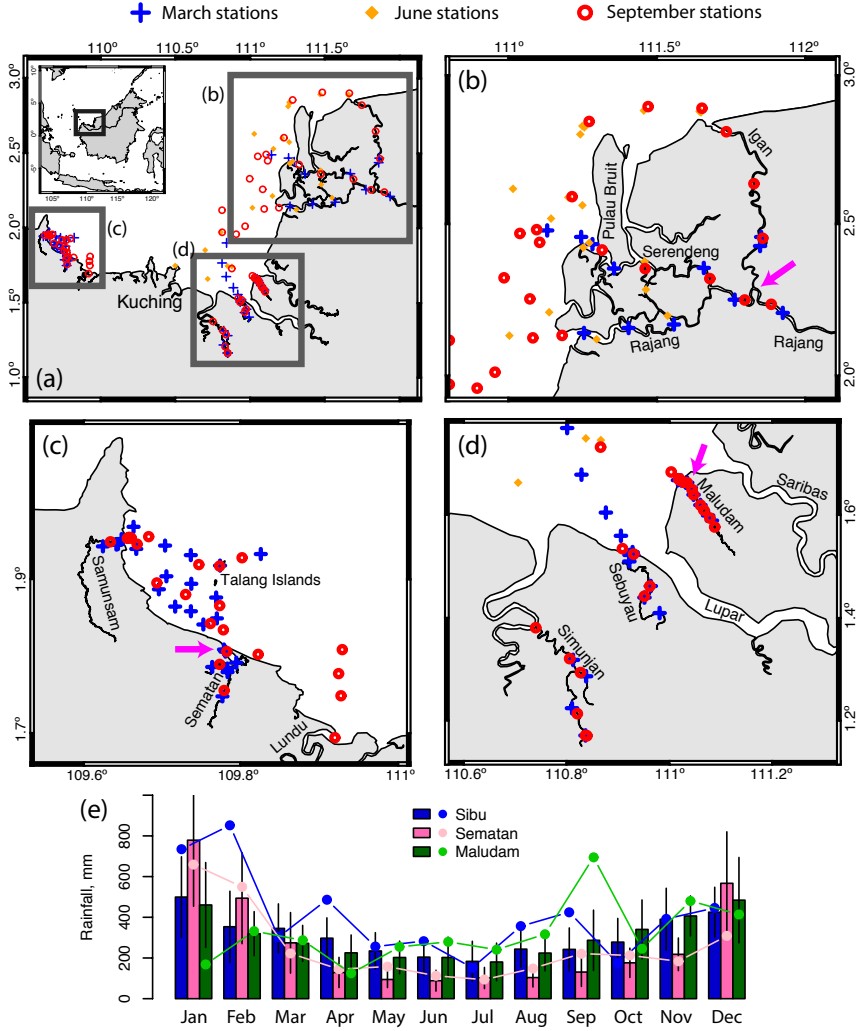

**Figure 1:** (a) Map of the study region showing station locations for each of the three expeditions. Thick grey boxes with letters indicate the areas shown in panels (b–d). (e) Monthly mean precipitation for the towns of Sibu, Sematan, and Maludam (marked with pink arrows in panels b, c, and d, respectively). Bars show mean ± 1 SD for 1999–2017, while points show values for 2017. Bars and points for the three locations in each month are separated horizontally for better readability, but correspond to the same time periods.





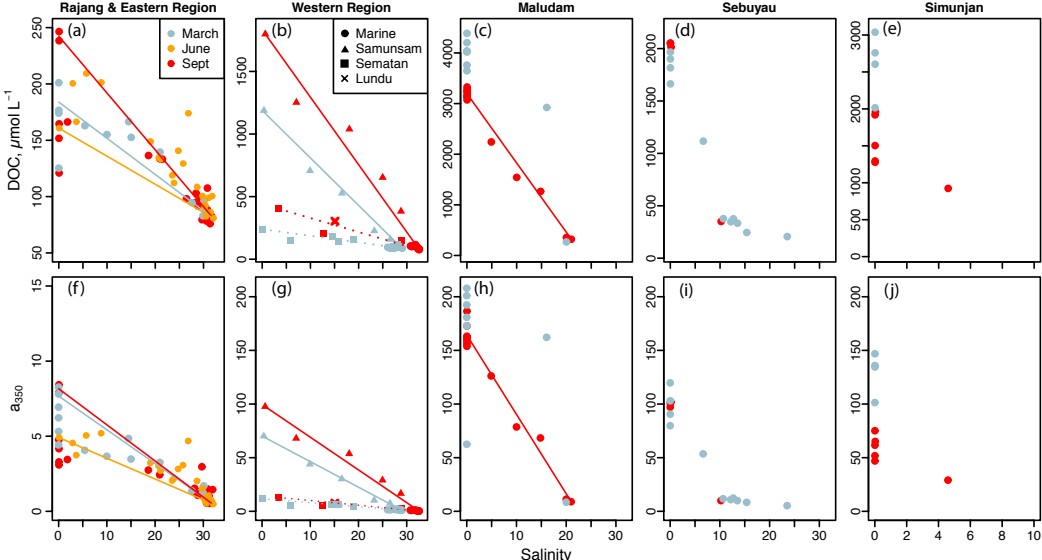

**Figure 2:** Changes in (a–e) dissolved organic carbon concentration, and (f–j) $a_{350}$ from rivers to coastal seawater. Coloured lines show conservative mixing models for the data from the corresponding season. In (b) and (c), solid *versus* dashed lines distinguish the mixing models for the Sematan and Samunsam rivers in the Western Region. Data are separated by sampling region in columns, indicated by the column titles. Colours of plotting symbols are used to distinguish sampling seasons in all panels as per the legend in panel (a).





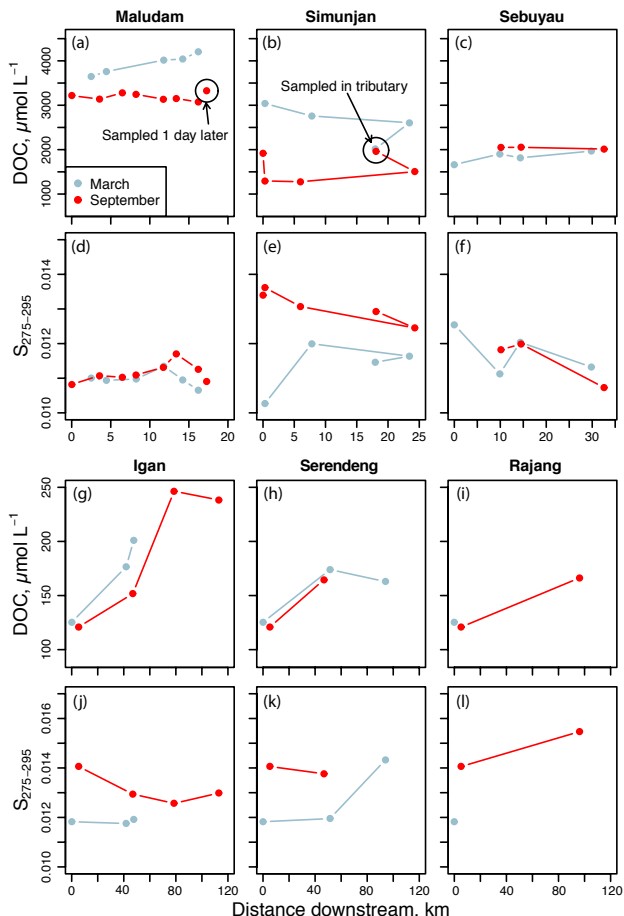

**Figure 3**: Changes in (a–c and g–i) dissolved organic carbon concentration and (d–f and j–l) $S_{275-295}$ with distance downstream for all stations with salinity of 0. Data in (a–f) are for the Maludam, Simunjan, and Sebuyau rivers, while (g–l) show data for the three main Rajang distributaries (named in Figure 1b): panels (h,k) show data for the Serendeng branch (includes the Lebaan and Paloh sections), while panels (i,l) show data for the Rajang branch (includes the Payang section).





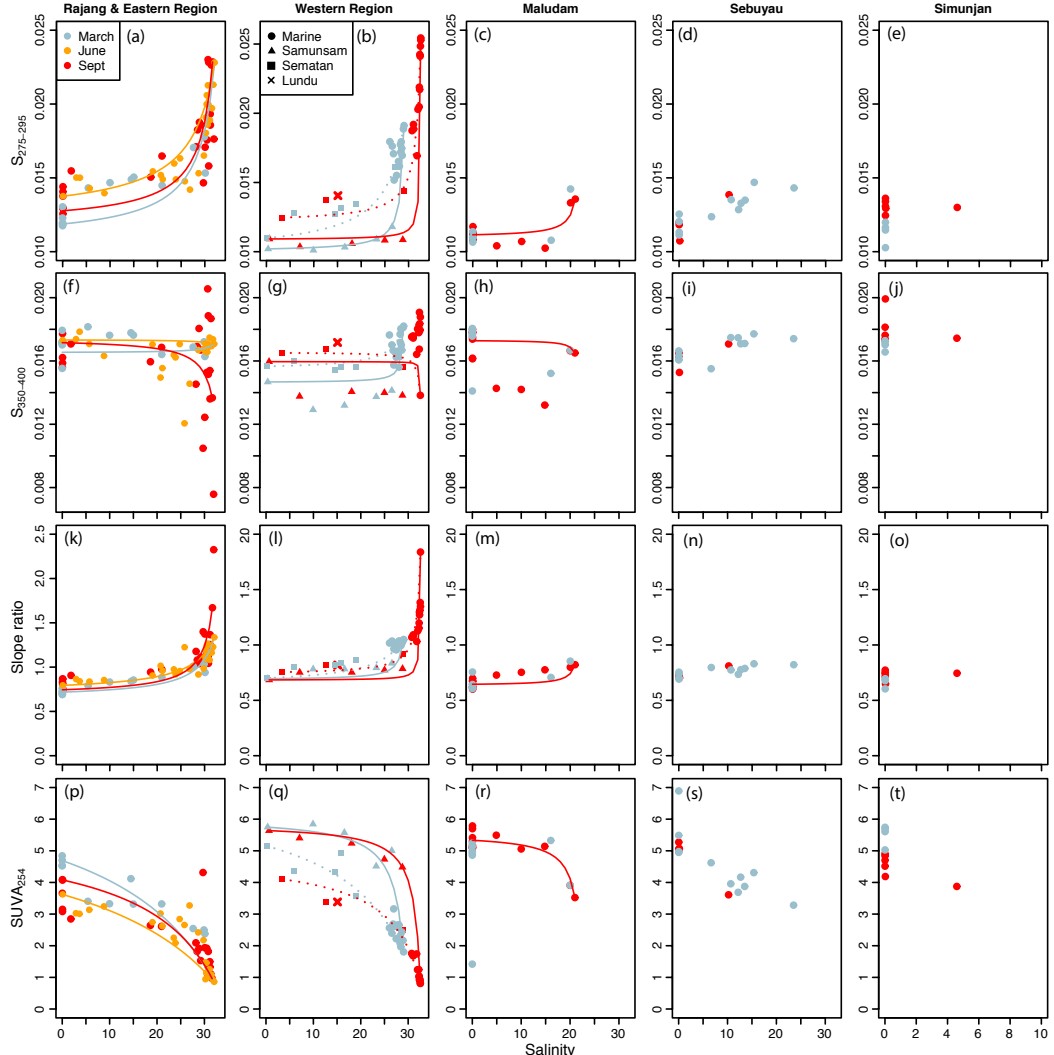

**Figure 4**: Changes in (a–e) $S_{275-295}$, (f–j) $S_{350-400}$, (k–o) CDOM spectral slope ratio, and (p–t) SUVA$_{254}$ from rivers to coastal seawater. Conservative mixing lines are shown as in Figure 2 (note that conservative mixing of CDOM properties is non-linear). Data are shown separately for each sampling region as indicated by column titles.





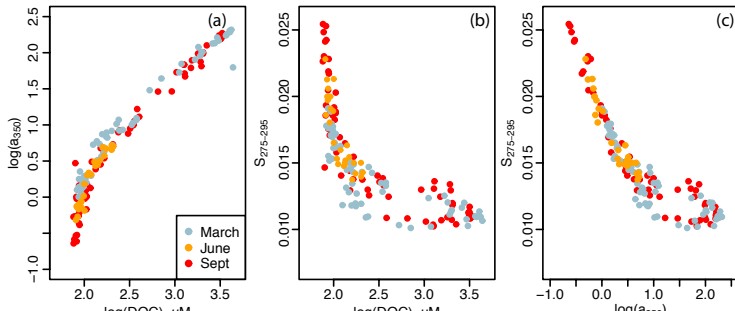

**Figure 5**: Scatter plots of (a) CDOM absorption *versus* DOC concentration, (b) $S_{275-295}$ *versus* DOC concentration, and (c) $S_{275-295}$ *versus* CDOM absorption for the entire dataset. Strong relationships were found between these parameters, but without seasonal variation.



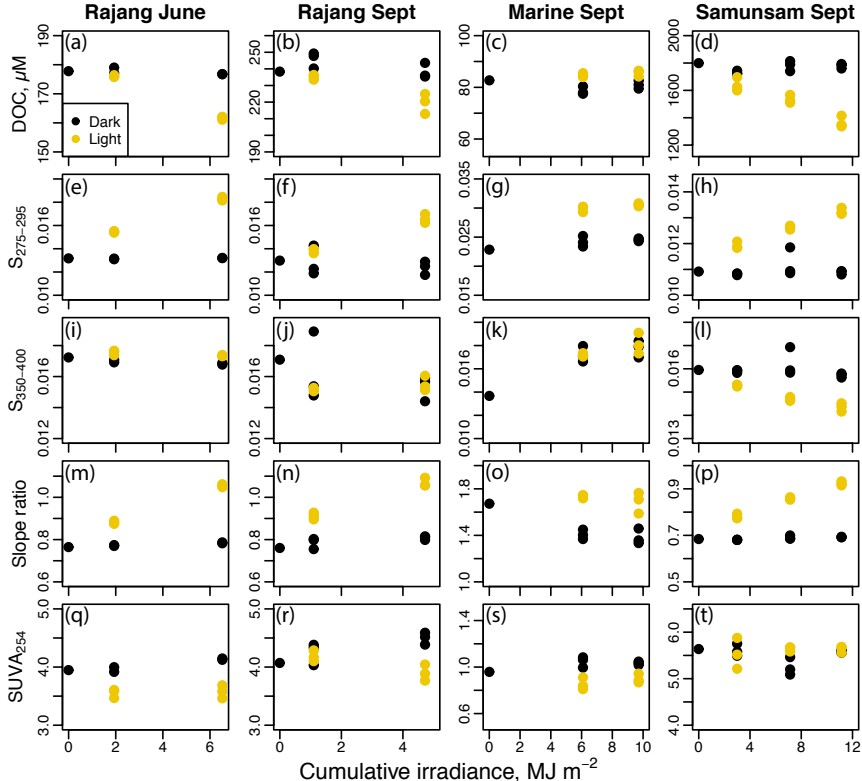

**Figure 6**: Results from photo-degradation experiments showing the decrease in DOC (top row), $S_{275-295}$ (second row), $S_{350-400}$ (third row), CDOM spectral slope ratio (fourth row), and $SUVA_{254}$ (bottom row) with cumulative irradiance from 318–450 nm wavelength. Each column corresponds to one degradation experiment, as indicated in the column titles.





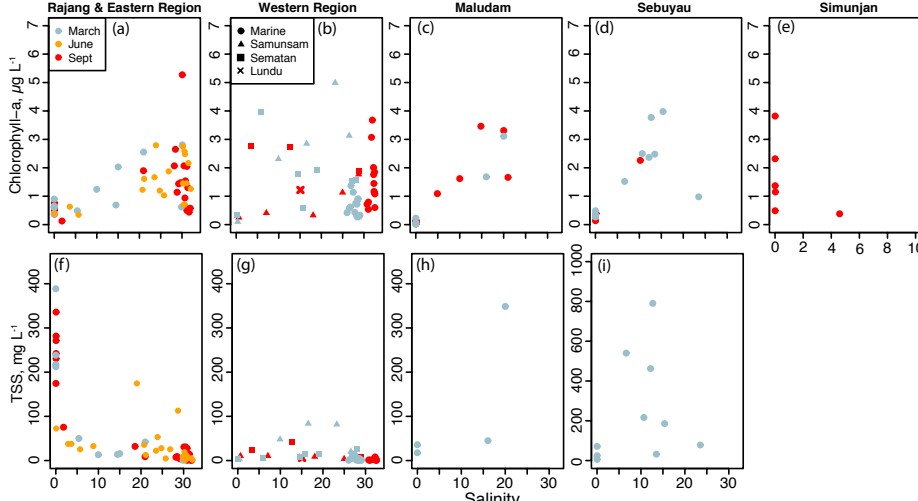

**Figure 7**: Distribution of (a–e) chlorophyll-*a*, and (f–i) total suspended solids from rivers to coastal seawater for each study region. TSS was not measured in the Simunjan.