# Peer review of "Distribution and cycling of terrigenous dissolved organic carbon in peatland-draining rivers and coastal waters of Sarawak, Borneo"

_Biogeosciences, 2018_

## Referee Comment (RC1) · Anonymous Referee #1 · 17 Sep 2018

Martin et al. present a concise empirical study on the spatial and temporal cycling of DOC in peatland draining rivers, Sarawak. Of particular interest are the results of the photo-degradation experiments and the fate of the riverine DOC component. The manuscript is well written and the study makes a valuable contribution to the scientific knowledge database. I have a few minor comments:

Page 5 Ln 30: acidification was chosen to preserve the DOC samples, however, this has been shown to reduce DOC concentrations (Kaplan.1994: https://doi.org/10.4319/lo.1994.39.6.1470) as well as alter spectral properties (Tfaily et al. 2011: doi:10.1016/j.aca.2011.08.037). I just wondered why cold storage was not

considered?

Page 6 Ln 2: Mentions that freezing did not affect the DOC results, how as this assessed? Is there any evidence that could be added (maybe to supplementary material to support?)

Page 8 Ln 9: were the quartz bottles overfilled with the water and then capped to eliminate headspace and therefore eliminate atmospheric exchange during the photo-degradation experiment? If so it would be good to include this information in the methods and also how the quartz bottles were prepared e.g. combusted? I wondered if there was any sign of bacterial growth in the water samples during the experiment (just out of curiosity).

Page 9 Ln 14: is there any way to include some information about the total distances travelled during the different campaigns (maybe on figure 1, by including a scale bar) or the distance between sampling stations to be added into the supplementary material? How was the distance between the sampling points decided?

Page 9 Ln 16: Was there any seasonal variability in salinity i.e. did you see a drop during the wet season due to greater freshwater input? Did you observe any salinity induced flocculation in the DOC samples which may have complicated analysis?

Page 11 Ln 26: Was there any evidence of photo-bleaching?

Page 13 Ln 14: Was any POC data taken? Would have been interesting to see if there was any change in the DOC:POC ratio during the photo-degradation experiment, but as stated POC to DOC turnover is unlikely to have been captured within the experiments time frame.

Page 13 Ln 24: Do you have any information on the extent of oil palm plantation coverage in this region/ the % of land likely to be covered by disturbed peatlands?

Page 13 Ln 27: Another reference that might be good to add is Materic et al. (2017) (https://doi.org/10.1038/s41598-017-16256-x) who observed differences in the composition of organic compounds between forest and disturbed peatlands (some data is from Sarawak).

Page 14 Ln 9: I wonder if the lower DOC concentrations observed at the end of the drier season could be a reflection of the interaction between the hydrology and photo-degradation i.e. during the dry season flow conditions will be low which could lead to longer residence times leading to greater UV exposure and thus DOC degradation. Again, just a thought.

Page 14 Ln 20: considering how photo liable the DOC is could this not have produced molecular level changes complicating its identification with respect to terrestrial sources, especially in samples collected further downstream which have been exposed to greater periods of light exposure? So maybe the terrigenous signal identified in the samples is even stronger? Just a thought.

Page 16 Ln 16: If the majority of the landscape is oil palm plantation I guess the drainage channels/ continuous yearly peat drainage could be ensuring that there is a continuous and direct flow of water (and thus DOC) into the river system, topping up the supply. Maybe this could also explain why there are higher DOC concentrations above the mixing line?

Page 16 Ln 19: As the majority of Rajang catchment is draining disturbed landcover (i.e. oil palm plantation) this could be contributing to an increased nutrient input from the pesticides and fertilisers and thus cause eutrophication to come degree and contribute to the DOC pool (even though I see that the chlorophyll is low). However, the sampling campaigns are designed to show us a 'snap-shot' of the spatial variability of the stream network across different seasons, so perhaps it is hard to rule out autochthonous DOC sources completely? Maybe there could be a lag response?

---

## Referee Comment (RC2) · Anonymous Referee #2 · 1 Oct 2018

Martin et al. explore the spatiotemporal variations of dissolved organic carbon (DOC) and colored dissolved organic matter (CDOM) using in-situ data obtained from a total of six peatland draining rivers and coastal zones in Sarawak. The photo-liability and the cycling of these riverine DOM are also further investigated and discussed by conducting field photo-degradation experiments. Although some of phenomena and conclusions presented in this work are not new in this region or elsewhere, it's valuable to have seasonally-resolved DOC measurements in this important tropical marine biodiversity hotspot area. Overall, the data obtained in "black waters" are very interesting and the overall quality of the study is positive and contributes to a better understanding of the DOM properties in a region that accounts for a large fraction of DOC export

to the global ocean while also facing strong anthropogenic influences; however, the manuscript needs some improvements in the text and figures.

Specific comments and suggestions for further improvements: The authors spent 5-pages to describe the Materials and Methods part, which is much longer than the results. Some of the methods described are not new (e.g., section 2.2.1 and 2.2.2 for measuring DOC concentration and CDOM absorbance), and can be properly shortened. Regarding the precipitation data in Fig. 1a, you may consider to highlight the locations of meteorological stations using corresponding colors in Fig. 1b-1d.

Regarding the section 2.2.3 of conservative mixing model, you may consider giving more details about the procedures using a table or other means.

Page 6, line 30, can you add a reference here and describe a bit the advantages of adding NaN3 to DI water as blank?

Page 7, line14, absorbance should have no unit. Also, page 7, line 1 – NaN3 absorbances around 26 m-1 at 230 nm, 4 m-1 at 254 nm: shouldn't these be absorption coefficients? These are very large values and likely to influence SUVA values.

Page 11, line 21-25, can you describe more about the highly scattered data in range of 2.5-3.5 of log DOC (Fig. 5b) and 1.0-2.0 of log S275-295 (Fig 5c) (e.g., geolocation of the scattered data) since you mentioned that no strong seasonal changes in DOM composition within your study region in discussions; Page 16, line 17-18), which contradict a bit with the results here. In addition, you may consider to keep same scale of these parameters in Fig. 5 instead of using log scale for some parameters.

Regarding Fig. 6, can you specify the black and yellow symbols in figure caption? Also, page 11, line 27, it is written that both "...DOC and CDOM decreasing after sunlight exposure" however, Figure 6 and Table 1 do not show CDOM absorption values. It would be important to also show the relative decrease in CDOM at 350 nm with light exposure. Figure 6 is also repeated in supplementary S3.

Page 13, line 10-15, it might not accurate to rule out a major autochthonous source of DOC just according to low surface Chl a concentration since the DOC fluxes from benthic flora to the overlying water column might also be another possible reason for high DOC in shallow estuary.

Page 14, line 10-14 "the high precipitation in Maludam in September ...…. DOC concentrations" which is not clear to me and I am wondering if there is previous study that reported this phenomenon; if yes, you may want to add a reference here. In addition, the lower DOC concentrations mentioned there could probably be associated with other environmental factors. The hydrological and meteorological conditions during wet and dry season could be different, which will change the residence time of waters and solar radiation, further affect DOC and CDOM properties there.

The authors may consider mentioning the conservative mixing model in the abstract and conclusions since the model was used to "validate" their DOC and CDOM measurements in the result for several times. In addition, it's better to describe the advantages or reasons to include this model in this work, so far, it looks weakly linked to other parts.

Regarding Table 1, please keep the font size and typeface consistent and change "*" to "×".

It would also be valuable to provide more information on the six rivers regarding their size, length, drainage basin, discharge, etc.

---

## Author Comment (AC2) · 25 Oct 2018

We thank Reviewer 2 for their time in reviewing our manuscript, and for providing constructive criticism. We are confident that we can revise our manuscript in a way that will address all of their questions satisfactorily. Our point-by point response is shown below, with the reviewer's comments quoted first, followed by our response.

Reviewer 2: Martin et al. explore the spatiotemporal variations of dissolved organic carbon (DOC) and colored dissolved organic matter (CDOM) using in-situ data obtained from a total of six peatland draining rivers and coastal zones in Sarawak. The photo-liability and the cycling of these riverine DOM are also further investigated and

discussed by conducting field photo-degradation experiments. Although some of phenomena and conclusions presented in this work are not new in this region or elsewhere, it's valuable to have seasonally-resolved DOC measurements in this important tropical marine biodiversity hotspot area. Overall, the data obtained in "black waters" are very interesting and the overall quality of the study is positive and contributes to a better understanding of the DOM properties in a region that accounts for a large fraction of DOC export to the global ocean while also facing strong anthropogenic influences; however, the manuscript needs some improvements in the text and figures. Specific comments and suggestions for further improvements: The authors spent 5- pages to describe the Materials and Methods part, which is much longer than the results. Some of the methods described are not new (e.g., section 2.2.1 and 2.2.2 for measuring DOC concentration and CDOM absorbance), and can be properly shortened. Regarding the precipitation data in Fig. 1a, you may consider to highlight the locations of meteorological stations using corresponding colors in Fig. 1b-1d.

Response: We agree that the methods were quite long. We will shorten the relevant sections, although because we need to describe numerous different analyses and the photo-degradation experiment, the Methods section will inevitably remain somewhat long. The meteorological stations were previously highlighted with pink arrows in each of the panels. We plan to change the arrow colour to match the colour of the bars in panel (e).

Reviewer 2: Regarding the section 2.2.3 of conservative mixing model, you may consider giving more details about the procedures using a table or other means.

Response: It's not clear to us exactly what additional details the reviewer is requesting. We will add an additional column to our supplementary data table to indicate which stations were used as end-members for the mixing models. We will also work on the wording in this section to make it clearer. We had already referenced the Stedmon & Markager paper that explains how CDOM mixing models should be calculated, and we plan to emphasise this further as a methods paper more clearly. The actual calculation
of two-endmember conservative mixing models is simple and is done very commonly in studies of estuarine gradients, so we do not want to expand the methods section even further by explaining this in detail. Basically, it is just calculating a weighted average concentration according to the proportion of each end-member in the mixture, so if freshwater and salinity 30 water are mixed to yield salinity 15, then the expected DOC concentration would be the average of the two end-member samples.

Reviewer 2: Page 6, line 30, can you add a reference here and describe a bit the advantages of adding NaN3 to DI water as blank?

Response: There is no specific reference for this, but maybe the reviewer misunderstood this slightly: we did not add NaN3 to the DI water in the reference cuvette of the spectrophotometer, we made proper reagent blanks with the NaN3 that were measured against the DI reference and then subtracted from the samples. We necessarily had to do this, because NaN3 does have some absorbance at wavelengths >250 nm, so we needed to correct for this. We will clarify this further in Section 2.2.2, and we also plan to add a supplementary figure to show a sodium azide blank spectrum.

Reviewer 2: Page 7, line14, absorbance should have no unit. Also, page 7, line 1 – NaN3 absorbances around 26 m-1 at 230 nm, 4 m-1 at 254 nm: shouldn't these be absorption coefficients? These are very large values and likely to influence SUVA values.

Response: We apologise for this oversight, these values are decadic absorption coefficients, not absorbances. We will correct the text. We report these numbers as decadic rather than Napierian absorption coefficients, because the sodium azide blank is most relevant for calculating SUVA254, which is done using decadic instead of Napierian coefficients. For the most part, the blanks at 254 nm were still small relative to the CDOM absorbance, given the high CDOM concentrations in most of our samples. Because the NaN3 concentration was also identical across all samples, this blank could be subtracted very accurately. On Page 11, we show that SUVA254 is extremely closely

related with SUVA at 280 nm, a wavelength at which NaN3 has no significant absorbance, across our dataset (r2 = 0.990). This indicates that our SUVA254 estimates were not affected by the NaN3 blank. We plan to show the spectrum of a sodium azide blank as a supplementary figure, which will help to illustrate this more clearly. We used NaN3 because this is the recommended preservation protocol for CDOM samples in the ocean colour / remote sensing community (as in Tilstone et al. 2001, cited on Page 5), and we are currently using our CDOM data for a satellite remote sensing analysis. However, because CDOM measurements for remote sensing purposes are usually made at wavelengths above 300 nm, we were unaware of this blank issue in advance. We then continued with this protocol for the sake of consistency across our dataset. With hindsight, it is maybe better not to use NaN3 for CDOM analysis at wavelengths below 300 nm, but we were clearly able to correct for this blank without significantly compromising our data.

Reviewer 2: Page 11, line 21-25, can you describe more about the highly scattered data in range of 2.5-3.5 of log DOC (Fig. 5b) and 1.0-2.0 of log S275-295 (Fig 5c) (e.g., geolocation of the scattered data) since you mentioned that no strong seasonal changes in DOM composition within your study region in discussions; Page 16, line 17-18), which contradict a bit with the results here. In addition, you may consider to keep same scale of these parameters in Fig. 5 instead of using log scale for some parameters.

Response: The scatter in these relationships at the high DOC concentrations (>2.5 log(DOC)) is largely due to inherent variability between the rivers: the Rajang, Sematan, and Simunjan have somewhat higher S275–295 at a given DOC or CDOM concentration than the Maludam or Samunsam. The lack of seasonal variation in these relationships is most obvious when comparing the March and September data: both datasets cover nearly the full range of data and both follow the same trajectory (including the scatter at higher DOC concentrations). The reason why it looks like there is some seasonality is two-fold: first, because we were unable to sample all sites in
all seasons, and thus the June data (only Rajang and marine samples) only cover a smaller range of values than the other seasons. Secondly, in June and September, we were able to sample marine waters with lower DOC concentrations, which could not be sampled because of weather conditions in March. Therefore, the March data do not extend to such low DOC / high S275–295 values as the other two seasons. However, our point here is that all of the data essentially follow the same trajectory on these plots, rather than clustering into two parallel relationships by season (as found in some studies of other regions). We plan to expand this section slightly to explain some of this in more detail, and hopefully this is enough to clarify our point sufficiently. We did try plotting these data using linear scales instead of log scales. Unfortunately, because our data span such a large range in DOC and in CDOM concentration, plotting on a linear scale makes it very hard to properly see most of the data, because it is not possible to distinguish properly any samples with less than about 250 $\mu$M DOC. We agree that log-scales make it harder to directly compare these data to our other figures, but otherwise the relationships we are trying to visualise are simply impossible to see across the full dataset.

Reviewer 2: Regarding Fig. 6, can you specify the black and yellow symbols in figure caption? Also, page 11, line 27, it is written that both ". . .DOC and CDOM decreasing after sunlight exposure" however, Figure 6 and Table 1 do not show CDOM absorption values. It would be important to also show the relative decrease in CDOM at 350 nm with light exposure. Figure 6 is also repeated in supplementary S3.

Response: We will specify the symbol colours in the figure caption. CDOM did indeed decrease, and we plan to add one more row of panels to Figure 6 to show the decrease in CDOM, as a350. We agree with the reviewer that showing the a350 data for the photodegradation experiments is important, and we therefore also plan to show the a350 data in Table 1 instead of showing the S350–400 data, given than S350–400 mostly showed little change relative to the controls, as is obvious in Figure 6. Unfortunately, the table is too small to show all these parameters. Figure 6 is not exactly repeated in

the supplementary information: our SI Fig. 3 shows the photo-degradation data plotted against the number of days of sunlight exposure, while Figure 6 shows the data plotted against our estimated cumulative irradiance. We think it is important to show both, because the different days do have differences in irradiance. Conversely, because we could not measure the cumulative irradiance exactly, we think it is important to also show the data plotted against time for comparison.

Reviewer 2: Page 13, line 10-15, it might not accurate to rule out a major autochthonous source of DOC just according to low surface Chl a concentration since the DOC fluxes from benthic flora to the overlying water column might also be another possible reason for high DOC in shallow estuary.

Response: This is a good point. Essentially, this would require significant stocks of benthic macrophytes in the rivers. However, although we have no systematic data of macrophyte cover, we believe that it is very unlikely that macrophytes are present in significant quantities in any of our rivers. First, in the Rajang and Sematan rivers, the suspended sediment concentrations are very high and prevent deep light penetration; the same is true for all of the blackwater rivers due to the high CDOM concentrations. Secchi depths were measured at some stations in the Rajang and were always less than 30 cm relative to a river depth of often $\geq$10 m. Moreover, although moderate to large amounts of terrestrial plant debris (branches and leaves of trees, tree trunks, entire clusters of palm trees) were always seen floating at the surface of all rivers and out at sea, we never observed debris of aquatic macrophytes. Exposed river banks at low tide also never showed evidence of aquatic macrophytes. In the blackwater rivers we often had the opportunity to see the upper 10–30 cm of the river bank below the water line, but also never saw any macrophytes, only the sediment. We will slightly expand this passage to explain that we never saw evidence of aquatic macrophytes, and that benthic primary production is likely to be at most minimal owing to the low light penetration in all rivers.

Reviewer 2: Page 14, line 10-14 "the high precipitation in Maludam in September ... .

.. DOC concentrations" which is not clear to me and I am wondering if there is previous study that reported this phenomenon; if yes, you may want to add a reference here. In addition, the lower DOC concentrations mentioned there could probably be associated with other environmental factors. The hydrological and meteorological conditions during wet and dry season could be different, which will change the residence time of waters and solar radiation, further affect DOC and CDOM properties there.

Response: We agree that this was not very clearly phrased. We will expand this section slightly to clarify our meaning: especially in peatlands, it has been noted that high precipitation can lower DOC concentrations in rivers by essentially creating a dilution effect, as described in the Clark et al. (2007) paper that we cited a few sentences previously. We will explain more clearly that this could be a dilution effect. Of course, the reviewer is quite correct in pointing out that numerous other environmental factors could influence the DOC seasonality in each river, and our intention was not to try and argue for one factor over another. However, because the Maludam catchment did appear to experience particularly high precipitation shortly before we sampled (as is apparent in Fig. 1), we feel that it is appropriate to point out that this might have influenced the apparent seasonality we recorded in this particular river.

Reviewer 2: The authors may consider mentioning the conservative mixing model in the abstract and conclusions since the model was used to "validate" their DOC and CDOM measurements in the result for several times. In addition, it's better to describe the advantages or reasons to include this model in this work, so far, it looks weakly linked to other parts.

Response: We already implicitly refer to the mixing models in the abstract when we state that "DOC and CDOM showed conservative mixing with seawater". Because the abstract is already quite long, it is probably better not to discuss the mixing models here in more detail. Calculating conservative mixing models is basically a standard practice when studying biogeochemical fluxes from rivers into seawater across the estuarine mixing zone. We never actually use the result of our mixing models to validate

our results, we use them instead to show whether or not our results are consistent with DOC and CDOM mixing conservatively, or whether non-conservative addition/removal is happening in the estuaries. This can't be done properly without actually calculating the mixing models. Given that it is an important objective of our study to determine whether or not the DOC and CDOM are mixing conservatively in these estuaries, the mixing models are actually integral to our analysis, which is why we show the curves of all our mixing models in Figures 2 and 4. These models essentially show us the theoretically expected changes in DOC and CDOM with salinity if all of the DOC and CDOM are mixing fully conservatively. In most cases we find that our data are consistent with these theoretical predictions, except in the Rajang. These conclusions cannot be properly supported without showing the theoretical mixing lines from the mixing models.

Reviewer 2: Regarding Table 1, please keep the font size and typeface consistent and change "*" to "×".

Response: This will be done.

Reviewer 2: It would also be valuable to provide more information on the six rivers regarding their size, length, drainage basin, discharge, etc.

Response: We will add some additional information about the rivers to Section 2.1, especially the approximate lengths of the rivers. Other manuscripts that are currently in preparation for this special issue will present more detailed information about the catchments, including estimates of the extent of peat soils and plantations in the catchments. Unfortunately, there are no readily available datasets for the areal extent of most of the drainage basins, the exact proportion of peat soils in each basin, or river discharge. These estimates are still being put together by other groups, so the information will ultimately be available within the special issue, but the data are not yet finalised enough to be summarised here. Since the objective of our manuscript is mostly to understand the distribution of DOC and CDOM across the river-to-seawater gradients and examine the biogeochemical processing of this DOM, this information is not critical for an understanding of our paper.

---

## Author Response (AR2)

**Authors' Response**

We are grateful to both of the reviewers and the associate editor for their constructive comments on our manuscript. We have revised the manuscript according to the points raised by both reviewers, and have also lightly edited the text to further improve the language and to shorten the text where possible. Our response to each of the reviewers is listed below.

**Response to Reviewer 1**

We thank Reviewer 1 for their time in reviewing our manuscript, and for providing constructive criticism. We have made a number of changes based on these comments, as detailed below (the reviewer's comments are quoted in blue italic font).

*Martin et al. present a concise empirical study on the spatial and temporal cycling of DOC in peatland draining rivers, Sarawak. Of particular interest are the results of the photo-degradation experiments and the fate of the riverine DOC component. The manuscript is well written and the study makes a valuable contribution to the scientific knowledge database. I have a few minor comments:*

*Page 5 Ln 30: acidification was chosen to preserve the DOC samples, however, this has been shown to reduce DOC concentrations (Kaplan.1994: https://doi.org/10.4319/lo.1994.39.6.1470) as well as alter spectral properties (Tfaily et al. 2011: doi:10.1016/j.aca.2011.08.037). I just wondered why cold storage was not considered?*

Appropriate sample storage is a problematic subject, to say the least, and neither for DOC nor for CDOM has any one protocol really become established above all others. In acidifying our DOC samples immediately upon collection and then storing them cold, we followed a method that is very commonly used in the oceanographic community, with acidification inhibiting microbial DOC metabolism. The Kaplan paper shows interesting results, although we note that it does not show a decrease in DOC upon acidification in all samples. Their analysis was also done with persulfate oxidation, which is typically less efficient than high-temperature combustion systems, so some of this apparent DOC loss could involve modification of DOC, not necessarily only mineralisation to $CO_2$. Given that we needed to store our samples for up to 1.5 months before they could be analysed, we decided that acid-preservation seemed like a safer approach than simple cold-storage; unfortunately, we didn't have the capacity to collect and store extra samples to test this.

We certainly agree that spectral properties of DOM would be altered upon acidification. All of our CDOM samples were therefore not acidified, but preserved with sodium azide to inhibit microbial activity, and stored cold alongside our DOC samples. As also explained in our response to Reviewer 2, we collected our CDOM data in part also to develop a remote sensing

algorithm, and this preservation method is the recommended protocol in the ocean colour community – although measurements are then generally made at wavelengths >300 nm. Given our experience with the absorbance blanks from NaN₃ up to around 270 nm, we agree with the reviewer than simple unpreserved cold storage is perhaps preferable if CDOM analysis in the UV range is planned (that said, as explained in the manuscript text and also in response to Reviewer 2, we are very confident that our blank correction was sufficiently accurate that this did not compromise our data). We have also added a new supplementary figure (now Supplementary Fig. 1) to show the absorption spectrum of the sodium azide blanks.

*Page 6 Ln 2: Mentions that freezing did not affect the DOC results, how as this assessed? Is there any evidence that could be added (maybe to supplementary material to support?)*

This was based on comparing the unfrozen samples to frozen samples from adjacent stations in the Maludam. Since all of the September Maludam samples follow very clear conservative estuarine mixing, with only a narrow range of DOC and CDOM parameters in freshwaters, frozen and unfrozen samples clearly had very similar results. Some protocols for DOC and for CDOM or FDOM samples in fact even recommend preservation by freezing, and while we think it is better not to alter samples in such a drastic way, major impacts from sample freezing are probably not common. We have added a brief explanation to this section to explain why we think that freezing did not impact these results.

*Page 8 Ln 9: were the quartz bottles overfilled with the water and then capped to eliminate headspace and therefore eliminate atmospheric exchange during the photo- degradation experiment? If so it would be good to include this information in the meth- ods and also how the quartz bottles were prepared e.g. combusted? I wondered if there was any sign of bacterial growth in the water samples during the experiment (just out of curiosity).*

The bottles were actually filled with a headspace of around 20–40 mL air, which was done to ensure that the samples would remain oxygenated throughout the duration of the experiment. If samples are filled without a headspace, DOC degradation can in principle become oxygen-limited, depending on the degradation rate. The quartz bottles were not combusted, but were acid-rinsed instead. This information has been added to the methods. We did not enumerate microbial cells in these samples, but there was no visible evidence of growth (no cloudiness or particulate matter visible).

*Page 9 Ln 14: is there any way to include some information about the total distances travelled during the different campaigns (maybe on figure 1, by including a scale bar) or the distance between sampling stations to be added into the supplementary material? How was the distance between the sampling points decided?*

We have added a scale bar to Figure 1a to give a quick overview of the distance. The other panels already contain a lot of information so we do not want to clutter them with further scale bars, but distances can be estimated easily from the latitude axes on these panels. Distances between river stations are partly shown in Figure 3 for those rivers where multiple freshwater samples were taken.

Distances between sampling points were inevitably decided according to multiple factors: during estuarine sampling we chose stations mostly according to salinity, but distance between stations was considered as well. Logistical considerations were naturally important, e.g. the total distance that could be covered in one day. Many of the marine sampling stations were chosen according to optical water types, because of the need to use our data for remote sensing purposes – in this case we selected stations according to distance to the coast, the presence of distinct fronts, and differences in water colour. Logistical considerations were of course again

important, e.g. sea conditions and prevailing currents, the total distance that could be covered given fuel, water, and time constraints, and the location of sand banks that had to be avoided for safety reasons.

*Page 9 Ln 16: Was there any seasonal variability in salinity i.e. did you see a drop during the wet season due to greater freshwater input? Did you observe any salinity induced flocculation in the DOC samples which may have complicated analysis?*

There was some seasonality in salinity in the Western Region, as mentioned at the end of Section 3.1 (this is also visible in Figures 2 and 4). We did not specifically test for salinity-induced flocculation of DOC owing to logistical constraints; we hope to test for this during future fieldwork in the region. However, none of the estuaries showed evidence of non-conservative removal of DOC, so we suspect that any salinity-induced flocculation probably does not have a significant impact on DOC concentrations in this region.

*Page 11 Ln 26: Was there any evidence of photo-bleaching?*

Yes. As also requested by Reviewer 2, we have now added additional panels to Figure 6 and Table 1 to show the decrease in CDOM concentration as $a_{350}$.

*Page 13 Ln 14: Was any POC data taken? Would have been interesting to see if there was any change in the DOC:POC ratio during the photo-degradation experiment, but as stated POC to DOC turnover is unlikely to have been captured within the experiments time frame.*

We did not measure POC either in our environmental samples or in the photodegradation samples. POC data for many of the stations will be presented in another manuscript in this special issue, but unfortunately the volume of water in the photodegradation incubations was insufficient anyway to filter for POC at each time-point. However, we are planning to do more follow-up work on photo-degradation of SE Asian peatland DOC, and we will certainly attempt to measure POC as part of that. Note that the water for the photo-degradation experiments was already filtered at the start of the experiment, and the incubation bottles were then sub-sampled at each time-point. These sub-samples were not re-filtered, so any DOC that was transformed to suspended POC would still be quantified as DOC with our measurements.

*Page 13 Ln 24: Do you have any information on the extent of oil palm plantation coverage in this region/ the % of land likely to be covered by disturbed peatlands?*

At this point we do not have updated estimates for plantation coverage and land disturbance for the region, but these questions are being addressed by other manuscripts that are in preparation for this special issue. Unfortunately, land-use data are difficult to obtain in this region on a catchment-basis. Essentially, practically all of the peatlands need to be considered as at least partly disturbed, in the sense that they have a history of logging, and even the Maludam National Park peat forest is classed as a secondary forest. In the Rajang delta in particular, the majority of peatlands has been converted to oil palm plantations in the past years. However, the objective of this paper is not specifically to analyse effects of land-use change on DOC and CDOM concentrations, and our sampling scheme was not designed specifically to address this question.

*Page 13 Ln 27: Another reference that might be good to add is Materic et al. (2017) (https://doi.org/10.1038/s41598-017-16256-x) who observed differences in the composition of organic compounds between forest and disturbed peatlands (some data is from Sarawak).*

Thanks, we had not yet come across this paper. We have added the citation.

*Page 14 Ln 9: I wonder if the lower DOC concentrations observed at the end of the drier season could be a reflection of the interaction between the hydrology and photo- degradation i.e. during the dry season flow conditions will be low which could lead to longer residence times leading to greater UV exposure and thus DOC degradation. Again, just a thought.*

We certainly cannot rule out this possibility, but we suspect that photo-degradation in the rivers is probably not so pronounced. All rivers showed significant flow in all seasons, and because of the extremely high light attenuation in all rivers (due to CDOM and sediments) a large change in water residence time would be needed to allow for significant photo-degradation within the rivers. Moreover, if photo-degradation was significant in the rivers, we would probably see more consistent seasonal variation in DOC across the rivers. Given that we can really only speculate about this, we have not made any changes to the manuscript in this regard.

*Page 14 Ln 20: considering how photo liable the DOC is could this not have produced molecular level changes complicating its identification with respect to terrestrial sources, especially in samples collected further downstream which have been exposed to greater periods of light exposure? So maybe the terrigenous signal identified in the samples is even stronger? Just a thought.*

Yes, this is a very valid point: we agree that any estimate of terrigenous contribution in marine samples based on $S_{275-295}$ would, if anything, be an under-estimate. As also mentioned above, we suspect that photo-degradation is not really at play within the rivers themselves, because of the short water residence times and the extremely high light attenuation. Instead, photo-degradation of tDOM most likely only becomes significant in coastal waters once sediments have partly settled and the euphotic depth is greater. We have not modified the manuscript further in this regard.

*Page 16 Ln 16: If the majority of the landscape is oil palm plantation I guess the drainage channels/ continuous yearly peat drainage could be ensuring that there is a continuous and direct flow of water (and thus DOC) into the river system, topping up the supply. Maybe this could also explain why there are higher DOC concentrations above the mixing line?*

This is certainly a possibility, although given the generally high year-round precipitation, it is likely that there is also simply a high amount of natural input year-round. The peatlands in the Rajang delta are drained naturally by a network of small rivers and streams that discharge into the Rajang distributary branches. We cannot say whether conversion to oil palm plantations may have increased this DOC input, although we cite the literature from site-specific studies elsewhere on Borneo that have shown increased DOC losses due to land-use. We have therefore decided not to make any changes to the manuscript in this regard, given that our focus in this paper is not to discuss the effects of land-use. Any discussion of natural *versus* plantation-induced DOC fluxes here would necessarily be entirely speculative.

*Page 16 Ln 19: As the majority of Rajang catchment is draining disturbed landcover (i.e. oil palm plantation) this could be contributing to an increased nutrient input from the pesticides and fertilisers and thus cause eutrophication to come degree and contribute to the DOC pool (even though I see that the chlorophyll is low). However, the sampling campaigns are designed to show us a 'snap-shot' of the spatial variability of the stream network across different seasons, so perhaps it is hard to rule out autochthonous DOC sources completely? Maybe there could be a lag response?*

Trying to understand the impact of fertiliser input from the plantations would be an interesting and important study. Depending on the magnitude of this nutrient input, it certainly might contribute to eutrophication. Again, however, we strongly suspect that primary production within the rivers is too strongly light-limited for there to be a response to nutrient input within

the rivers and estuaries, so most likely there would be a down-stream effect in coastal waters, once the water clarity is higher. The question of how the underwater light level is controlled in this region will be addressed in a separate manuscript that is currently in preparation for this special issue. We would point out that although our sampling is indeed a snap-shot, any nutrient input from the plantations should have been happening also well in advance of our sampling campaign, so if there really was a downstream increase in chlorophyll due to nutrient inputs we would expect to see elevated chlorophyll concentrations within the rivers and estuaries, but the chlorophyll concentrations there are mostly very low. However, a more thorough discussion of these patterns will be the subject of the bio-optical manuscript that is currently in preparation.

Response to Reviewer 2

We thank Reviewer 2 for their time in reviewing our manuscript, and for providing constructive criticism. We have made a number of changes to the manuscript based on these comments, and our point-by-point responses are detailed below (the reviewer's comments are quoted in blue italic font).

*Martin et al. explore the spatiotemporal variations of dissolved organic carbon (DOC) and colored dissolved organic matter (CDOM) using in-situ data obtained from a total of six peatland draining rivers and coastal zones in Sarawak. The photo-liability and the cycling of these riverine DOM are also further investigated and discussed by conducting field photo-degradation experiments. Although some of phenomena and conclusions presented in this work are not new in this region or elsewhere, it's valuable to have seasonally-resolved DOC measurements in this important tropical marine biodiversity hotspot area. Overall, the data obtained in "black waters" are very interesting and the overall quality of the study is positive and contributes to a better understanding of the DOM properties in a region that accounts for a large fraction of DOC export to the global ocean while also facing strong anthropogenic influences; however, the manuscript needs some improvements in the text and figures.*

*Specific comments and suggestions for further improvements: The authors spent 5- pages to describe the Materials and Methods part, which is much longer than the results. Some of the methods described are not new (e.g., section 2.2.1 and 2.2.2 for measuring DOC concentration and CDOM absorbance), and can be properly shortened. Regarding the precipitation data in Fig. 1a, you may consider to highlight the locations of meteorological stations using corresponding colors in Fig. 1b-1d.*

We agree that the methods were quite long. We have shortened the relevant sections, although because we need to describe numerous different analyses and the photo-degradation experiment, the Methods section is inevitably somewhat long.

The meteorological stations were previously highlighted with pink arrows in each of the panels. We have now changed the arrow colour to match the colour of the bars in panel (e).

*Regarding the section 2.2.3 of conservative mixing model, you may consider giving more details about the procedures using a table or other means.*

It's not clear to us exactly what additional details the reviewer is requesting. We have decided to add an additional column to our supplementary data table that indicates which stations were used as end-members for the mixing models. We have also changed the wording in this section a little to make it clearer. We had already referenced the Stedmon & Markager paper that explains how CDOM mixing models should be calculated, and we now emphasise this as a methods paper more clearly. The actual calculation of two-endmember conservative mixing models is simple and this is done very commonly in studies of estuarine gradients, so we do not want to expand the methods section even further by explaining this in detail.

*Page 6, line 30, can you add a reference here and describe a bit the advantages of adding NaN3 to DI water as blank?*

There is no specific reference for this, but maybe the reviewer misunderstood this slightly: we did not add $NaN_3$ to the DI water in the reference cuvette of the spectrophotometer, we made proper reagent blanks with the $NaN_3$ that were measured against the DI reference and then subtracted from the samples. We necessarily had to do this, because $NaN_3$ does have some absorbance at wavelengths >250 nm, so we needed to correct for this. We have tried to clarify this as best as possible in Section 2.2.2 without adding excessive length.

*Page 7, line14, absorbance should have no unit. Also, page 7, line 1 – NaN3 absorbances around 26 m-1 at 230 nm, 4 m-1 at 254 nm: shouldn't these be absorption coefficients? These are very large values and likely to influence SUVA values.*

We apologise for this oversight, these values are decadic absorption coefficients, not absorbances. We have corrected the text. We report these as decadic rather than Napierian absorption coefficients here because the blank is most relevant for calculating $SUVA_{254}$, which is done using decadic instead of Napierian coefficients. For the most part, the blanks at 254 nm were still small relative to the CDOM absorbance, given the high CDOM concentrations in most of our samples. Because the $NaN_3$ concentration was also identical across all samples, this blank could be subtracted very accurately. On Page 11, we show that $SUVA_{254}$ is extremely closely related with SUVA at 280 nm, a wavelength at which $NaN_3$ has no significant absorbance, across our dataset ($r^2$ = 0.990). This indicates that our $SUVA_{254}$ estimates were not affected by the $NaN_3$ blank. We have now also added a new supplementary figure (now Supplementary Fig. 1) to show the spectrum of a sodium azide blank.

We used $NaN_3$ because this is the recommended preservation protocol for CDOM samples in the ocean colour / remote sensing community (as in Tilstone et al. 2001, cited on Page 5), and we are currently using our CDOM data for a satellite remote sensing analysis. However, because CDOM measurements for remote sensing purposes are usually made at wavelengths above 300 nm, we were unaware of this blank issue in advance. We then continued with this protocol for the sake of consistency across our dataset. With hindsight, it is maybe better not to use $NaN_3$ for CDOM analysis at wavelengths below 300 nm, but we were clearly able to correct for this blank without significantly compromising our data.

*Page 11, line 21-25, can you describe more about the highly scattered data in range of 2.5-3.5 of log DOC (Fig. 5b) and 1.0-2.0 of log S275-295 (Fig 5c) (e.g., geolocation of the scattered data) since you*

*mentioned that no strong seasonal changes in DOM composition within your study region in discussions; Page 16, line 17-18), which contradict a bit with the results here. In addition, you may consider to keep same scale of these parameters in Fig. 5 instead of using log scale for some parameters.*

The scatter in these relationships at the high DOC concentrations (>2.5 log(DOC)) is largely due to inherent variability between the rivers: the Rajang, Sematan, and Simunjan have somewhat higher $S_{275-295}$ at a given DOC or CDOM concentration than the Maludam or Samunsam. The lack of seasonal variation in these relationships is most obvious when comparing the March and September data: both datasets cover nearly the full range of data and both follow the same trajectory (including the scatter at higher DOC concentrations). The reason why it looks like there is some seasonality is two-fold: first, because we were unable to sample all sites in all seasons, and thus the June data (only Rajang and marine samples) only cover a smaller range of values than the other seasons. Secondly, in June and September, we were able to sample marine waters with lower DOC concentrations, which could not be sampled because of weather conditions in March. Therefore, the March data do not extend to such low DOC / high $S_{275-295}$ values as the other two seasons. However, our point here is that all of the data essentially follow the same trajectory on these plots, rather than clustering into two parallel relationships by season (as found in some studies of other regions).

We have expanded this section slightly to explain some of this in more detail, and hopefully this is enough to clarify our point sufficiently.

We did try plotting these data using linear scales instead of log-scales. Unfortunately, because our data span such a large range in DOC and in CDOM concentration, plotting on a linear scale makes it very hard to properly see most of the data, because it is not possible to distinguish properly any samples with less than about 250 µM DOC. We agree that log-scales make it harder to directly compare these data to our other figures, but otherwise the relationships we are trying to visualise are simply impossible to see across the full dataset.

*Regarding Fig. 6, can you specify the black and yellow symbols in figure caption? Also, page 11, line 27, it is written that both ". . .DOC and CDOM decreasing after sunlight exposure" however, Figure 6 and Table 1 do not show CDOM absorption values. It would be important to also show the relative decrease in CDOM at 350 nm with light exposure. Figure 6 is also repeated in supplementary S3.*

We have specified the symbol colours in the figure caption now. CDOM did indeed decrease, and we have now added one more row of panels to Figure 6 to show the decrease in CDOM, as $a_{350}$. We agree with the reviewer that showing the $a_{350}$ data for the photodegradation experiments is important, and we have therefore also decided to show the $a_{350}$ data in Table 1 instead of showing the $S_{350-400}$ data, given than $S_{350-400}$ mostly showed little change relative to the controls, as is obvious in Figure 6. Unfortunately, the table is too small to show all these parameters.

Figure 6 is not exactly repeated in the supplementary information: our SI Fig. 3 shows the photo-degradation data plotted against the number of days of sunlight exposure, while Figure 6 shows the data plotted against our estimated cumulative irradiance. We think it is important to show both, because the different days do have differences in irradiance. Conversely, because we could not measure the cumulative irradiance exactly, we think it is important to also show the data plotted against time for comparison.

*Page 13, line 10-15, it might not accurate to rule out a major autochthonous source of DOC just according to low surface Chl a concentration since the DOC fluxes from benthic flora to the overlying water column might also be another possible reason for high DOC in shallow estuary.*

This is a good point. Essentially, this would require significant stocks of benthic macrophytes in the river. However, although we have no systematic data of macrophyte cover, we believe that it is very unlikely that macrophytes are present in significant quantities in any of our rivers. First, in the Rajang and Sematan rivers, the suspended sediment concentrations are very high and prevent deep light penetration; the same is true for all of the blackwater rivers due to the high CDOM concentrations. Secchi depths were measured at some stations in the Rajang and were always less than 30 cm relative to a river depth of often ≥10 m. Moreover, although moderate to large amounts of terrestrial plant debris (branches and leaves of trees, tree trunks, entire clusters of palm trees) were always seen floating at the surface of all rivers and out at sea, we never observed debris of aquatic macrophytes. Exposed river banks at low tide also never showed evidence of aquatic macrophytes. In the blackwater rivers we often had the opportunity to see the upper 10–30 cm of the river bank below the water line, but also never saw any macrophytes, only the sediment. We have now slightly expanded this passage to state that we never saw evidence of aquatic macrophytes, and that benthic primary production is likely to be at most minimal owing to the low light penetration in all rivers.

*Page 14, line 10-14 "the high precipitation in Maludam in September ... . .. DOC concentrations" which is not clear to me and I am wondering if there is previous study that reported this phenomenon; if yes, you may want to add a reference here. In addition, the lower DOC concentrations mentioned there could probably be associated with other environmental factors. The hydrological and meteorological conditions during wet and dry season could be different, which will change the residence time of waters and solar radiation, further affect DOC and CDOM properties there.*

We agree that this was not very clearly phrased. We have expanded this section slightly to clarify our meaning: especially in peatlands, it has been noted that high precipitation can lower DOC concentrations in rivers by essentially creating a dilution effect, as described in the Clark et al. (2007) paper that we cited a few sentences previously. We now refer to the Clark paper again in this sentence and explain that this could be a dilution effect. Of course, the reviewer is quite correct in pointing out that numerous other environmental factors could influence the DOC seasonality in each river, and our intention was not to try and argue for one factor over another (hopefully this is now clearer in the manuscript). However, because the Maludam catchment did appear to experience particularly high precipitation shortly before we sampled (as is apparent in Fig. 1), we feel that it is appropriate to point out that this might have influenced the apparent seasonality we recorded in this particular river.

*The authors may consider mentioning the conservative mixing model in the abstract and conclusions since the model was used to "validate" their DOC and CDOM measurements in the result for several times. In addition, it's better to describe the advantages or reasons to include this model in this work, so far, it looks weakly linked to other parts.*

We already implicitly refer to the mixing models in the abstract when we state that "DOC and CDOM showed conservative mixing with seawater". Because the abstract is already around 275 words, we have decided not to include more information here.

Calculating conservative mixing models is basically a standard practice when studying biogeochemical fluxes from rivers into seawater across the estuarine mixing zone. We never actually use the result of our mixing models to validate our results, we use them instead to show whether or not our results are consistent with DOC and CDOM mixing conservatively, or whether non-conservative addition/removal is happening in the estuaries. This can't be done properly without actually calculating the mixing models. Given that it is an important objective of our study to determine whether or not the DOC and CDOM are mixing conservatively in these estuaries, the mixing models are actually integral to our analysis, which is why we show the curves of all our mixing models in Figures 2 and 4. These models essentially show us the theoretically expected changes in DOC and CDOM with salinity if all of the DOC and CDOM are mixing fully conservatively. In most cases we find that our data are consistent with these theoretical predictions, except in the Rajang. These conclusions cannot be properly supported without showing the theoretical mixing lines from the mixing models.

*Regarding Table 1, please keep the font size and typeface consistent and change "\*" to "×".*

Done

*It would also be valuable to provide more information on the six rivers regarding their size, length, drainage basin, discharge, etc.*

We have added some additional information about the rivers to Section 2.1, especially the approximate lengths of the rivers. Other manuscripts that are currently in preparation for this special issue will present more detailed information about the catchments, including estimates of the extent of peat soils and plantations in the catchments. Unfortunately, there are no readily available datasets for the areal extent of most of the drainage basins, the exact proportion of peat soils in each basin, or river discharge. These estimates are still being put together by other groups, so the information will ultimately be available within the special issue, but the data are not yet finalised enough to be summarised here. Since the objective of our manuscript is mostly to understand the distribution of DOC and CDOM across the river-to-seawater gradients and examine the biogeochemical processing of this DOM, this information is clearly not critical for an understanding of our paper.

**Distribution and cycling of terrigenous dissolved organic carbon in peatland-draining rivers and coastal waters of Sarawak, Borneo**

Patrick Martin[1], Nagur Cherukuru[2], Ashleen S. Y. Tan[1*], Nivedita Sanwlani[1], Aazani Mujahid[3], Moritz Müller[4]

[1]Asian School of the Environment, Nanyang Technological University, Singapore 639798, Singapore
[2]CSIRO Oceans and Atmosphere Flagship, Canberra ACT 2601, Australia
[3]Department of Aquatic Science, Faculty of Resource Science & Technology, University Malaysia Sarawak, 94300 Kota Samarahan, Sarawak, Malaysia
[4]Swinburne University of Technology, Faculty of Engineering, Computing and Science, 93350 Kuching, Sarawak, Malaysia

*Correspondence to*: Patrick Martin (pmartin@ntu.edu.sg)

*Current address: Erasmus Mundus Joint Programme in Marine Environment and Resources, Plentzia Marine Station, University of the Basque Country, Plentzia, Spain.

[revised manuscript text omitted]
 March, June, and September 2017. Six rivers were sampled in March and September: the Rajang (~550 km length), the Maludam (~33 km length), the Sebuyau (~58 km length), the Simunjan (~54 km length), the Sematan (~15 km length), and the Samunsam (~34 km length) (Fig. 1). The June expedition sampled only the Rajang river. On all expeditions, the river estuaries and open coastal waters were sampled (Fig. 1). In September, one sample was also taken in the estuary of a seventh river, the Lundu river (94 km length). All station locations, sampling dates, and measured data are shown in Supplementary Table 1. Four of the rivers (Maludam, Simunjan, Sebuyau, and Samunsam) are blackwater rivers that drain catchments with high peatland coverage, while the Sematan and Lundu rivers

drain catchments with limited peatland cover. The Rajang river drains mineral soils until the town of Sibu, from where it branches into multiple distributary channels (Fig. 1). The distributaries each have unique names; the main ones (Rajang, Serendeng, and Igan) are identified in Fig. 1. These distributaries are surrounded by extensive peatlands that drain directly into the distributary channels (Staub et al., 2000). Mangroves grow along the estuaries of all of the rivers. All river samples are distinguished below by river name, while marine samples are distinguished by whether they were collected in the region east of Kuching ("Eastern Region", influenced strongly by the Rajang river outflow), or in the region west of Kuching ("Western Region", influenced by the Samunsam and Sematan rivers). The Talang Islands in the Western Region (Fig. 1) are surrounded by coral reefs.

The three sampling periods corresponded to the end of the north-east monsoon (March, end of the wettest season of the year), the south-west monsoon (June, lower precipitation), and shortly before the beginning of the north-east monsoon (September, end of the drier season). Monthly precipitation across Sarawak can vary several-fold across the year, but is mostly ≥100 mm per month (Sa'adi et al., 2017). Precipitation data were obtained from weather stations in Sibu, Maludam, and Sematan. Monthly averages were calculated for the period 1999–2017, omitting the few months for Maludam and Sematan for which there were days with missing data (there were no missing data in 2017). Precipitation in 2017 was mostly within 1 standard deviation of the 1999–2017 means (Fig. 1e). It should be noted that precipitation in this region is strongly driven by small-scale convective systems; however, 2017 was overall not an unusual year in terms of precipitation. Water temperatures in Sarawak show essentially no seasonal variation (average water temperatures during all expeditions fell within 28.5–29.5º C).

To collect samples in the Rajang river and the Eastern Region, a liveaboard fishing boat was chartered for 4–7 day cruises, and all samples were filtered and preserved upon collection. All other stations were sampled from small outboard-powered boats, in which case samples were stored dark at ambient temperature in insulated boxes on board, and filtered back on land each afternoon/evening. All samples were collected within the upper 1 m using either a bucket or a hand-held jug; sampling devices were rinsed thoroughly with sample water before sampling. Sample water was decanted into either amber borosilicate glass bottles (DOC and CDOM) or HDPE bottles (chlorophyll and total suspended solids).

DOC and CDOM samples were filtered through 0.2-μm pore-size Anodisc filters (47-mm diameter) using an all-glass filtration system that was rinsed with 1 M HCl and ultrapure deionised water (18.2 MΩ cm$^{-1}$, referred to as "DI water" below) in between each sample. Each Anodisc filter was rinsed by filtering 100–150 mL of DI water and then 50–100 mL of sample water, before a further 100–150 mL of sample water were filtered and taken as the sample. DOC samples (30 mL) were immediately acidified with 100 μL of either 25% $H_3PO_4$ (March expedition) or 50% $H_2SO_4$ (all other samples) to pH <2.0. CDOM samples (30 mL) were preserved with 150 μL of 10 g L$^{-1}$ $NaN_3$, following Tilstone et al. (2001). DOC and CDOM samples were stored in amber borosilicate vials with PTFE-lined septa at +4° C until analysis (within 1.5 months of collection), although some river samples in September froze for 1-2 days due to a refrigerator malfunction in the field. However, freezing did not appear to affect the DOC or CDOM results, as seen from comparing DOC and CDOM data for samples from adjacent stations in the Maludam river that did and did not freeze.

Deleted: , the Sibu precipitation record does not represent the entire Rajang river catchment; rather, we take the precipitation data as showing that 2017 was overall not an unusual year in terms of precipitation

Samples (50–1000 mL) for chlorophyll-$a$ were filtered onto pre-ashed (450º C, 4 h) 25-mm diameter Whatman GF/F filters, wrapped in aluminium foil, and immediately frozen in a liquid nitrogen dry shipper. They were stored in the dry shipper until analysis within 6 months of collection.

Samples for total suspended solids (TSS, 50–1000 mL) were filtered onto pre-ashed (450º C, 4 h), pre-rinsed, pre-weighed 25-mm diameter Whatman GF/F filters. Filters were rinsed three times with DI water and stored at -20º C in Petri dishes. Procedural blanks for all parameters were prepared in the field using DI water.

**2.2 Chemical analyses**

**2.2.1 Dissolved organic carbon analysis**

Dissolved organic carbon was analysed as non-purgeable organic carbon on a Shimadzu TOC-L system with a high-salt combustion kit after a 5-min sparge, using potassium hydrogen phthalate for calibration. Instrument performance was monitored using certified Deep-Sea Water from the Hansell Laboratory, University of Miami (42–45 μmol L$^{-1}$). Our analyses consistently yielded slightly higher values for the reference water, with a long-term mean ± 1 SD of 47 ± 2.0 μmol L$^{-1}$ (n = 51). Procedural blanks prepared in the field almost all contained <10 μmol L$^{-1}$, except for those prepared in between blackwater river samples, which contained 13–27 μmol L$^{-1}$; a correction for these procedural blanks was not applied.

**2.2.2 Chromophoric dissolved organic matter analysis**

CDOM samples were warmed to room temperature and their absorbance measured from 230–900 nm against a DI water reference, using a Thermo Evolution 300 dual-beam spectrophotometer. Samples from March were measured in either a 10-cm or a 1-cm pathlength quartz cuvette, or in a 1-cm quartz cuvette after 10-fold dilution with DI water (for blackwaters). Samples from June and September were measured undiluted in either 10-cm, 1-cm, or 0.2-cm pathlength cuvettes. Instrument performance was checked according to Mitchell et al. (2000). Reagent blanks of NaN$_3$ in DI water were measured and subtracted from all samples. NaN$_3$ was found to absorb significantly from 230–265 nm, with decadic absorption coefficients of ~26 m$^{-1}$ at 230 nm, ~4 m$^{-1}$ at 254 nm, but ≤0.1 m$^{-1}$ at wavelengths ≥275 nm (Supplementary Fig. 1). Blank absorbances at wavelengths ≥275 nm were nearly always <10% of sample absorbances, and mostly around 1% or less. CDOM spectra were baseline-corrected (Green and Blough, 1994), and converted to Napierian absorption coefficients following Eq. (1):

$$a_\lambda = 2.303 \times \frac{A_\lambda}{l},\qquad(1)$$

where $a_\lambda$ and $A_\lambda$ are, respectively, the absorption coefficient and the absorbance at wavelength $\lambda$, and $l$ is the cuvette pathlength in m. These calculations were performed using the R package hyperSpec (Beleites and Sergo, 2018). Our raw CDOM spectra (as decadic absorption coefficients) are shown in Supplementary Table 1, and representative spectra are shown in Supplementary Fig. 2. CDOM spectral slope coefficients were calculated for the intervals 275–295 nm and 350–

Deleted: Carrier gas (80 mL min$^{-1}$) was produced by a zero-air generator. Analytical blanks were prepared freshly for each run using water from an Elga Purelab Flex 3 system (18.2 MΩ cm$^{-1}$, includes a UV lamp and TOC monitor); these blanks were identical to or lower than certified Low-Carbon Water from the Hansell Laboratory, University of Miami.
All

400 nm using linear regression of log-transformed data as in Helms et al. (2008). Specific UV Absorbance at 254 nm (SUVA$_{254}$) was calculated from the decadic absorption coefficient at 254 nm and DOC concentration in mg L$^{-1}$. Dissolved inorganic constituents such as bromide, sulphide, nitrate, iodide, and molecular oxygen have negligible absorbance at wavelengths ≳250 nm compared to CDOM (Fally et al., 2000; Guenther et al., 2001).

**2.2.3 Conservative mixing models for DOC and CDOM**

Two-endmember mixing models for DOC and CDOM were calculated for the Rajang, Samunsam, and Sematan rivers, and in September also for the Maludam river. For the other rivers/seasons, there was either insufficient variation in salinity (Maludam and Simunjan), or the salinity was influenced strongly by adjacent rivers that were not sampled (Sebuyau, which drains into the Lupar river estuary). Linear mixing models were calculated from the end-member DOC concentrations and CDOM spectra at salinity intervals of 1.0 from salinity 0 (river water) to the salinity of the marine end-member station (29–32.5). For CDOM, we calculated the full absorption spectrum at each interval and then re-calculated the spectral slopes and SUVA$_{254}$ values, following Stedmon and Markager (2003). It should be noted that conservative mixing of CDOM results in non-linear changes in spectral slopes and SUVA with salinity. Appropriate end-member stations were identified from salinity, DOC, and geographical location (Supplementary Table 1). In March, rough seas prevented us from sampling fully marine waters in the Eastern Region, so the marine end-member station from the June expedition was used instead.

**2.2.4 Chlorophyll-*a* and total suspended solids analysis**

Chlorophyll samples were extracted in 10 mL 90% acetone at -20º C in the dark for 24 h, and fluorescence measured at excitation 436 nm / emission 680 nm (both with 5 nm bandpass) on a Horiba Fluoromax 4 spectrofluorometer (Welschmeyer, 1994). The fluorescence signal was normalised to the excitation lamp reference intensity and calibrated against a chlorophyll-*a* standard from spinach (Sigma-Aldrich, C5753). The limit of detection (3 SD of the blank) was <2 ng chlorophyll per filter.

TSS samples were dried at 75º C for 24 h before re-weighing. In March and September, they were then ashed at 450º C for 1 hour to remove organic matter and weighed again to determine inorganic weight. All weighing was performed on a Mettler-Toledo microbalance with ±1 µg accuracy.

**2.3 Photo-degradation experiments**

Four short-term photo-degradation experiments were conducted in the field (one in June, three in September). For each, 1 L of sample water was filtered as for DOC and CDOM samples (using multiple Anodiscs if necessary), and filled into 150 mL acid-washed quartz bottles with ground quartz stoppers, leaving a headspace to prevent O$_2$ limitation. 
[revised manuscript text omitted]
). The increased scatter in Figs. 5b,c at high DOC and CDOM concentrations is due to the fact that the rivers differed somewhat in S$_{275-295}$: in particular, the Rajang, Sematan, and Simunjan had higher S$_{275-295}$ for a given DOC or CDOM concentration than the Samunsam and Maludam. There was no seasonal variation in any of these relationships, inasmuch as the datasets from all three seasons plot along a single trajectory in all three plots, rather than segregating into parallel trajectories by season.

**3.3 Photo-degradation of DOC and CDOM**

DOM from the Rajang and Samunsam rivers was photo-labile, with DOC and CDOM decreasing after sunlight exposure. In contrast, marine water collected in the Eastern Region only showed some changes in CDOM, but no decrease in DOC (Fig. 6, Table 1). Daily irradiances, integrated from 318–450 nm, ranged from 0.92 to 3.00 MJ m$^{-2}$, with cumulative irradiances for each experiment ranging from 5–11 MJ m$^{-2}$. Irradiance data for each day are shown in Supplementary Fig. 3. In practice, plotting our data against estimated cumulative irradiance showed the same trends as plotting simply against time of exposure (Supplementary Fig. 4), although we estimate that the Samunsam water received a slightly higher irradiance over five days than the marine water over six days, and that the two Rajang experiments differed by about 20% in irradiance despite both lasting three days.

 The Rajang water in June lost 16.1 ± 0.5 μmol L$^{-1}$ DOC by the end of the experiment (mean ± 1 SD, representing 8.8%–9.4% of the starting DOC), with $a_{350}$ decreasing as well. S$_{275-295}$ and S$_R$ both increased, while S$_{350-400}$ remained essentially unchanged, and SUVA$_{254}$ decreased slightly (Fig. 6). In September, we found very similar changes in the Rajang water after sunlight exposure: 18.9 ± 6.1 μmol L$^{-1}$ DOC were lost (mean ± 1 SD, representing 5.6%–10.7% of starting DOC), $a_{350}$ decreased, and S$_{275-295}$ and S$_R$ increased by amounts similar to June. Although S$_{350-400}$ decreased slightly relative to the initial sample, there was no difference between light and dark bottles in this parameter. SUVA$_{254}$ decreased slightly in the light

[revised manuscript text omitted]

**Rajang experiment September**

| Day | Cumulative irradiance | DOC, µmol L$^{-1}$ Light | DOC, µmol L$^{-1}$ Dark | $S_{275-295}$ Light | $S_{275-295}$ Dark | $a_{350}$ Light | $a_{350}$ Dark |
|---|---|---|---|---|---|---|---|
| 0 | 0 | 238 | | 0.0130 | | 7.91 | |
| 1 | $1.10\times10^6$ | 235 ± 1.4 | 246 ± 4.8 | 0.0138 ± 2.0×10$^{-4}$ | 0.0128 ± 1.3×10$^{-4}$ | 8.00 ± 0.08 | 8.89 ± 0.67 |
| 3 | $4.72\times10^6$ | 219 ± 6.1 | 238 ± 4.6 | 0.0166 ± 3.7×10$^{-4}$ | 0.0124 ± 5.8×10$^{-4}$ | 6.04 ± 0.25 | 9.24 ± 0.64 |

**Marine experiment September**

| Day | Cumulative irradiance | DOC, µmol L$^{-1}$ Light | DOC, µmol L$^{-1}$ Dark | $S_{275-295}$ Light | $S_{275-295}$ Dark | $a_{350}$ Light | $a_{350}$ Dark |
|---|---|---|---|---|---|---|---|
| 0 | 0 | 83 | | 0.0228 | | 0.55 | |
| 4 | $6.08\times10^6$ | 85 ± 0.8 | 79 ± 1.6 | 0.0299 ± 5.2×10$^{-4}$ | 0.0242 ± 9.1×10$^{-4}$ | 0.261 ± 0.022 | 0.402 ± 0.045 |
| 6 | $9.71\times10^6$ | 86 ± 1.2 | 81 ± 1.6 | 0.0306 ± 2.5×10$^{-4}$ | 0.0245 ± 2.3×10$^{-4}$ | 0.247 ± 0.006 | 0.385 ± 0.012 |

**Samunsam experiment September**

| Day | Cumulative irradiance | DOC, µmol L$^{-1}$ Light | DOC, µmol L$^{-1}$ Dark | $S_{275-295}$ Light | $S_{275-295}$ Dark | $a_{350}$ Light | $a_{350}$ Dark |
|---|---|---|---|---|---|---|---|
| 0 | 0 | 1799 | | 0.0109 | | 97.6 | |
| 1 | $2.99\times10^6$ | 1640 ± 50 | 1730 ± 11 | 0.0119 ± 1.3×10$^{-4}$ | 0.0108 ± 3.2×10$^{-5}$ | 85.0 ± 2.6 | 93.5 ± 1.5 |
| 3 | $7.13\times10^6$ | 1535 ± 30 | 1781 ± 38 | 0.0126 ± 7.0×10$^{-5}$ | 0.0112 ± 5.6×10$^{-4}$ | 79.9 ± 2.3 | 89.2 ± 5.9 |
| 5 | $11.2\times10^6$ | 1366 ± 42 | 1779 ± 17 | 0.0133 ± 1.2×10$^{-4}$ | 0.0109 ± 7.1×10$^{-5}$ | 70.0 ± 3.2 | 96.4 ± 1.2 |

[Figure]

**Figure 1:** (a) Map of the study region showing station locations for each of the three expeditions. Thick grey boxes with letters indicate the areas shown in panels (b–d). (e) Monthly mean precipitation for the towns of Sibu, Sematan, and Maludam (locations of the rain gauges are marked with arrows in panels b, c, and d; arrow colours correspond to the bar colours in panel e). Bars show mean ± 1 SD for 1999–2017, while points show values for 2017. Bars and points for the three locations in each month are separated horizontally for better readability, but correspond to the same time periods.

[Figure]

**Figure 2:** Changes in (a–e) dissolved organic carbon concentration, and (f–j) $a_{350}$ from rivers to coastal seawater. Coloured lines show conservative mixing models for the data from the corresponding season. In (b) and (c), solid *versus* dashed lines distinguish the mixing models for the Sematan and Samunsam rivers in the Western Region. Data are separated by sampling region in columns, indicated by the column titles. Colours of plotting symbols are used to distinguish sampling seasons in all panels as per the legend in panel (a).

[Figure]

**Figure 3**: Changes in (a–c and g–i) dissolved organic carbon concentration and (d–f and j–l) $S_{275\text{-}295}$ with distance downstream for all stations with salinity of 0. Data in (a–f) are for the Maludam, Simunjan, and Sebuyau rivers, while (g–l) show data for the three main Rajang distributaries (named in Figure 1b): panels (h,k) show data for the Serendeng branch (includes the Lebaan and Paloh sections), while panels (i,l) show data for the Rajang branch (includes the Payang section).

[Figure]

**Figure 4**: Changes in (a–e) $S_{275-295}$, (f–j) $S_{350-400}$, (k–o) CDOM spectral slope ratio, and (p–t) SUVA$_{254}$ from rivers to coastal seawater. Conservative mixing lines are shown as in Figure 2 (note that conservative mixing of CDOM properties is non-linear). Data are shown separately for each sampling region as indicated by column titles.

[Figure]

**Figure 5**: Scatter plots of (a) CDOM absorption *versus* DOC concentration, (b) $S_{275-295}$ *versus* DOC concentration, and (c) $S_{275-295}$ *versus* CDOM absorption for the entire dataset. Strong relationships were found between these parameters, but without seasonal variation.

[Figure]

**Figure 6**: Results from photo-degradation experiments showing the decrease in DOC (top row), CDOM concentration (second row), $S_{275-295}$ (third row), $S_{350-400}$ (fourth row), CDOM spectral slope ratio (fifth row), and $SUVA_{254}$ (bottom row) with cumulative irradiance from 318–450 nm wavelength. Each column corresponds to one degradation experiment, as indicated in the column titles. Black symbols indicate dark control bottles, yellow symbols indicate light-exposed bottles.

| Deleted: second |
| Deleted: third |
| Deleted: fourth |

[Figure]

**Figure 7**: Distribution of (a–e) chlorophyll-*a*, and (f–i) total suspended solids from rivers to coastal seawater for each study region. TSS was not measured in the Simunjan.